# Graph Neural Network with Local Frame for Molecular Potential Energy Surface

**Xiyuan Wang**[1,2], **Muhan Zhang**[1,3]
[1]Institute for Artificial Intelligence, Peking University
[2]School of Intelligence Science and Technology, Peking University
[3]Beijing Institute for General Artificial Intelligence
`wangxiyuan@pku.edu.cn,muhan@pku.edu.cn`

## Abstract

Modeling molecular potential energy surface is of pivotal importance in science. Graph Neural Networks have shown great success in this field. However, their message passing schemes need special designs to capture geometric information and fulfill symmetry requirement like rotation equivariance, leading to complicated architectures. To avoid these designs, we introduce a novel *local frame* method to molecule representation learning and analyze its expressivity. Projected onto a frame, equivariant features like 3D coordinates are converted to invariant features, so that we can capture geometric information with these projections and decouple the symmetry requirement from GNN design. Theoretically, we prove that given non-degenerate frames, even ordinary GNNs can encode molecules injectively and reach maximum expressivity with coordinate projection and frame-frame projection. In experiments, our model uses a simple ordinary GNN architecture yet achieves state-of-the-art accuracy. The simpler architecture also leads to higher scalability. Our model only takes about $30\%$ inference time and $10\%$ GPU memory compared to the most efficient baselines.

## 1 Introduction

Prediction of molecular properties is widely used in fields such as material searching, drug designing, and understanding chemical reactions [1]. Among properties, potential energy surface (PES) [2], the relationship between the energy of a molecule and its geometry, is of pivotal importance as it can determine the dynamics of molecular systems and many other properties. Many computational chemistry methods have been developed for the prediction, but few can achieve both high precision and scalability.

In recent years, machine learning (ML) methods have emerged, which are both accurate and efficient. Graph Neural Networks (GNNs) are promising among these ML methods. They have improved continuously [3–10] and achieved state-of-the-art performance on many benchmark datasets. Compared with popular GNNs used in other graph tasks [11], these models need special designs, as molecules are more than a graph composed of merely nodes and edges. Atoms are in the continuous 3D space, and the prediction targets like energy are sensitive to the coordinates of atoms. Therefore, GNNs for molecules must include geometric information. Moreover, these models should keep the symmetry of the target properties for generalization. For example, the energy prediction should be invariant to the coordinate transformations in O(3) group, like rotation and reflection.

All existing methods can keep the invariance. Some models [4, 5, 8] use hand-crafted invariant features like distance, angle, and dihedral angle as the input of GNN. Others use equivariant representations, which change with the coordinate transformations. Among them, some [6, 9, 12] use irreducible representations of the SO(3) group. The other models [7, 10] manually design functions for equivariant and invariant representations. All these methods can keep invariance, but they vary

X. Wang et al., Graph Neural Network with Local Frame for Molecular Potential Energy Surface. *Proceedings of the First Learning on Graphs Conference (LoG 2022)*, PMLR 198, Virtual Event, December 9–12, 2022.

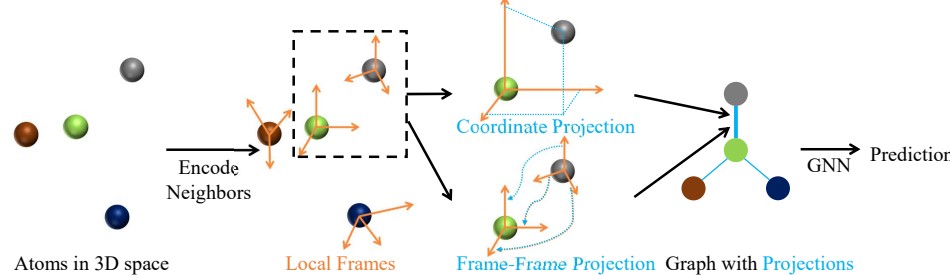

**Figure 1:** An illustration of our model. One local frame is generated for each atom. Frames are used to transform geometric information into invariant representations. Then an ordinary GNN is applied.

in performance. Therefore, expressivity analysis is necessary. However, the symmetry requirement hinders the application of the existing theoretical framework for ordinary GNNs [13].

By using the local frame, we decouple the symmetry requirement. As shown in Figure 1, our model, namely *GNN-LF*, first produces a frame (a set of bases of $\mathbb{R}^3$ space) equivariant to O(3) transformations. Then it projects the relative positions and frames of neighbor atoms on the frame as the edge features. Therefore, an ordinary GNN with no special design for symmetry can work on the graph with only invariant features. The expressivity of the GNN for molecules can also be proved using a framework for ordinary GNNs [13]. As the GNN needs no special design for symmetry, GNN-LF also has a simpler architecture and, thus, better scalability. Our model achieves state-of-the-art performance on the MD17 and QM9 datasets. It also uses only 30% time and 10% GPU memory than the fastest baseline on the PES task.

## 2 Preliminaries

**Ordinary GNN.** Message passing neural network (MPNN) [14] is a common framework of GNNs. For each node, a message passing layer aggregates information from neighbors to update the node representations. The $k^{\text{th}}$ layer can be formulated as follows.

$$\boldsymbol{h}_v^{(k)} = \mathrm{U}^{(k)}(\boldsymbol{h}_v^{(k-1)}, \sum_{u \in N(v)} M^{(k)}(\boldsymbol{h}_u^{(k-1)}, e_{vu})) \tag{1}$$

where $\boldsymbol{h}_v^{(k)}$ is the representations of node $v$ at the $k^{\text{th}}$ layer, $N(v)$ is the set of neighbors of $v$, $\boldsymbol{h}_v^{(0)}$ is the node $v$'s features, $e_{uv}$ is the features of edge $uv$, and $U^{(k)}, M^{(k)}$ are some functions.

Xu et al. [13] provide a theoretical framework for the expressivity of ordinary GNNs. One message passing layer can encode neighbor nodes injectively and then reaches maximum expressivity. With several message passing layers, MPNN can learn the information of multi-hop neighbors.

**Modeling PES.** PES is the relationship between molecular energy and geometry. Given a molecule with $N$ atoms, our model takes the kinds of atoms $z \in \mathbb{Z}^N$ and the 3D coordinates of atoms $\vec{r} \in \mathbb{R}^{N \times 3}$ as input to predict the energy $\hat{\mathcal{E}} \in \mathbb{R}$ of this molecule. It can also predict the force $\hat{\vec{\mathcal{F}}} \in \mathbb{R}^{N \times 3} = -\nabla_{\vec{r}} \hat{\mathcal{E}}$.

**Equivariance.** To formalized the symmetry requirement, we define equivariant and invariant functions as in [15].

**Definition 2.1.** *Given a function $h : \mathbb{X} \to \mathbb{Y}$ and a group $G$ acting on $\mathbb{X}$ and $\mathbb{Y}$ as $\star$. We say that $h$ is*

$$G\text{-invariant:} \quad if\, h(g \star x) = h(x), \;\; \forall x \in \mathbb{X}, g \in G \tag{2}$$

$$G\text{-equivariant:} \quad if\, h(g \star x) = g \star h(x), \;\; \forall x \in \mathbb{X}, g \in G \tag{3}$$

The energy is invariant to the permutation of atoms, coordinates' translations, and coordinates' orthogonal transformations (rotations and reflections). GNN naturally keeps the permutation invariance. As the relative position $\vec{r}_{ij} = \vec{r}_i - \vec{r}_j \in \mathbb{R}^{1 \times 3}$, which is invariant to translation, is used as the input to GNNs, the translation invariance can also be ensured. So we focus on orthogonal transformations. Orthogonal transformations of coordinates form the group $\mathrm{O}(3) = \{Q \in \mathbb{R}^{3 \times 3} \mid QQ^T = I\}$, where

$I$ is the identity matrix. Representations are considered as **functions of** $z$ **and** $\vec{r}$, so we can define equivariant and invariant representations.

**Definition 2.2.** *Representation $s$ is called an **invariant representation** if $s(z, \vec{r}) = s(z, \vec{r}o^T), \forall o \in O(3), z \in \mathbb{Z}^N, \vec{r} \in \mathbb{R}^{N \times 3}$. Representation $\vec{v}$ is called an **equivariant representation** if $\vec{v}(z, \vec{r})o^T = \vec{v}(z, \vec{r}o^T), \forall o \in O(3), z \in \mathbb{Z}^N, \vec{r} \in \mathbb{R}^{N \times 3}$.*

Invariant and equivariant representations are also called scalar and vector representations respectively in some previous work [7].

**Frame** is a special kind of equivariant representation. Through our theoretical analysis, frame $\vec{E}$ is an orthogonal matrix in $\mathbb{R}^{3 \times 3}, \vec{E}\vec{E}^T = I$. GNN-LF generates a frame $\vec{E}_i \in \mathbb{R}^{3 \times 3}$ for each node $i$. We will discuss how to generate the frames in Section 5.

In Lemma 2.1, we introduce some basic operations of representations.

**Lemma 2.1.**

- *Any function of invariant representation $s$ will produce an invariant representation.*

- *Let $s \in \mathbb{R}^F$ denote an invariant representation, $\vec{v} \in \mathbb{R}^{F \times 3}$ denote an equivariant representation. We define $s \cdot \vec{v} \in \mathbb{R}^{F \times 3}$ as a matrix whose $(i, j)$th element is $s_i \vec{v}_{ij}$. When $\vec{v} \in \mathbb{R}^{1 \times 3}$, we first broadcast it along the first dimension. Then the output is also an equivariant representation.*

- *Let $\vec{v} \in \mathbb{R}^{F \times 3}$ denote an equivariant representation. $\vec{E} \in \mathbb{R}^{3 \times 3}$ denotes an equivariant frame. The **projection** of $\vec{v}$ to $\vec{E}$, denoted as $P_{\vec{E}}(\vec{v}) := \vec{v}\vec{E}^T$, is an invariant representation in $\mathbb{R}^{F \times 3}$. For $\vec{v}$, $P_{\vec{E}}$ is a bijective function. Its **inverse** $P_{\vec{E}}^{-1}$ convert an invariant representation $s \in \mathbb{R}^{F \times 3}$ to an equivariant representation in $\mathbb{R}^{F \times 3}$, $P_{\vec{E}}^{-1}(s) = s\vec{E}$.*

- *Projection of $\vec{v}$ to a general equivariant representation $\vec{v}' \in \mathbb{R}^{F' \times 3}$ is an invariant representation in $\mathbb{R}^{F \times F'}$, $P_{\vec{v}'}(\vec{v}) = \vec{v}\vec{v}'^T$.*

**Local Environment.** Most PES models set a *cutoff radius* $r_c$ and encode the *local environment* of each atom as defined in Definition 2.3.

**Definition 2.3.** *Let $r_{ij}$ denote $||\vec{r}_{ij}||$. The **local environment** of atom $i$ is $LE_i = \{(s_j, \vec{r}_{ij}) | r_{ij} < r_c\}$, the set of invariant atom features $s_j$ (like atomic numbers) and relative positions $\vec{r}_{ij}$ of atoms $j$ within the sphere centered at $i$ with cutoff distance $r_c$, where $r_c$ is usually a hyperparameter.*

In this work, orthogonal transformation of a set/sequence means transforming each element in the set/sequence. For example, an orthogonal transformation $o$ will map $\{(s_j, \vec{r}_{ij}) | r_{ij} < r_c\}$ to $\{(s_j, \vec{r}_{ij}o^T) | r_{ij} < r_c\}$.

## 3 Related work

We classify existing ML models for PES into two classes: manual descriptors and GNNs. GNN-LF outperforms the representative of each kind in experiments.

**Manual Descriptor.** These models first use manually designed functions with few learnable parameters to convert one molecule to a descriptor vector and then feed the vector into some ordinary ML models like kernel regression [16–18] and neural network [19–21] to produce the prediction. These methods are more scalable and data-efficient than GNNs. However, due to the hard-coded descriptors, they are less accurate and cannot process variable-size molecules or different kinds of atoms.

**GNN.** These GNNs mainly differ in the way to incorporate geometric information.

*Invariant models* use rotation-invariant geometric features only. Schutt et al. [3] and Schütt et al. [4] only consider the distance between atoms. Klicpera et al. [5] introduce angular features, and Gasteiger et al. [8] further use dihedral angles. Similar to GNN-LF, the input of the GNN is invariant. However, the features are largely hand-crafted and are not expressive enough, while our projections on frames are learnable and provably expressive. Moreover, as some features are of multiple atoms (for example, angle is a feature of three-atom tuple), the message passing scheme passes messages between node tuples rather than nodes, while GNN-LF uses an ordinary GNN with lower time complexity.

Recent works have also utilized equivariant features, which will change as the input coordinates rotate. Some [6, 9, 12, 22] are based on *irreducible representations of the $SO(3)$ group*. Though having

certain theoretical expressivity guarantees [23], these methods and analyses are based on polynomial approximation. High-order tensors are needed to approximate complex functions like high-order polynomials. However, in implementation, only low-order tensors are used, and these models' empirical performance is not high. Other works [7, 10] model equivariant interactions in Cartesian space using both invariant and equivariant representations. They achieve good empirical performance but have no theoretical guarantees. Different sets of functions must be designed separately for different input and output types (invariant or equivariant representations), so their architectures are also complex. Our work adopts a completely different approach. We introduce O(3)-equivariant frames and project all equivariant features on the frames. The expressivity can be proved using the existing framework [13] and needs no high-order tensors.

**Frame models**. Some of existing methods [24, 25] designed for other tasks also use frame to decouple the symmetry requirement. However, in conclusion, these methods differ significantly from ours in task, theory, and method as follows.

- Most target properties of molecules are O(3)-equivariant or invariant (including energy and force). Our model can fully describe symmetry, while existing "frame" models cannot. For example, a molecule and its mirroring must have the same energy, and GNN-LF will produce the same prediction while existing models cannot keep the invariance.
- Our theoretical analysis removes group representation used in [23, 26].
- Existing models use some schemes not learnable to initialize frames and update them. GNN-LF uses a learnable message passing scheme to produce frames and will not update them, leading to simpler architecture and lower overhead.

The comparison is detailed in Appendix F.

## 4 How frames boost expressivity?

Though symmetry imposes constraints on our design, our primary focus is expressivity. Therefore, we only discuss how the frame boosts expressivity in this section. Our methods, implementations, and how our model keeps invariance will be detailed in Section 6 and Appendix J. We assume the existence of frames in this section and will discuss it in Section 5. All proofs are in Appendix A.

### 4.1 Decoupling symmetry requirement

Though equivariant representations have been used for a long time, it is still unclear how to transform them ideally. Existing methods [7, 10, 15, 27] either have no theoretical guarantee or tend to use too many parameters. This section asks a fundamental question: can we use invariant representations instead of equivariant ones and keep expressivity?

Given any frame $\vec{E}$, the projection $P_{\vec{E}}(\vec{x})$ will contain all the information of the input equivariant feature $\vec{x}$, because the inverse projection function can resume $\vec{x}$ from projection, $P_{\vec{E}}^{-1}(P_{\vec{E}}(\vec{x})) = \vec{x}$. Therefore, we can use $P_{\vec{E}}$ and $P_{\vec{E}}^{-1}$ to change the type (invariant or equivariant representation) of input and output of any function without information loss.

**Proposition 4.1.** *Given frame $\vec{E}$ and **any** equivariant function $g$, there exists a function $\tilde{g} = P_{\vec{E}} \circ g \circ P_{\vec{E}}^{-1}$ which takes invariant representations as input and outputs invariant representations, where $\circ$ is function composition. $g$ can be expressed with $\tilde{g}$: $g = P_{\vec{E}}^{-1} \circ \tilde{g} \circ P_{\vec{E}}$.*

We can use a multilayer perceptron (MLP) to approximate the function $\tilde{g}$ and thus achieving **universal approximation** for all O(3)-equivariant functions. Proposition 4.1 motivates us to transform equivariant representations to projections in the beginning and then fully operate on the invariant representation space. Invariant representations can also be transformed back to equivariant prediction with inverse projection operation if necessary.

### 4.2 Projection boosts message passing layer

The previous section discusses how projection decouples the symmetry requirement. This section shows that projections contain rich geometry information. Even ordinary GNNs can reach maximum expressivity with projections on frames, while existing models with hand-crafted invariant features

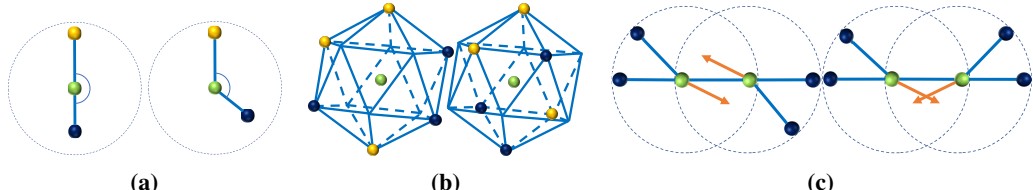

**Figure 2:** The green balls in the figure are the center atoms. We use balls with different colors to represent different kinds of atoms. (a) SchNet cannot distinguish two local environments due to the inability to capture angle. (b) DimeNet cannot distinguish two local environments with the same set of angles. Blue lines form a regular icosahedron and help visualization. The center atom is at the symmetrical center of the icosahedron. (c) Invariant models fail to pass the orientation information, while the projection of frame vectors can solve this problem. For simplicity, we only show one vector (orange) to represent the frame.

are not expressive enough. The discussion is composed of two parts. Coordinate projection boosts the expressivity of one single message passing layer, and frame-frame projection boosts the whole GNN composed of multiple message passing layers.

Note that in this section, we consider input $x_1$, $x_2$ (local environment or the whole molecule) as equal if they can interconvert with some orthogonal transformation ($\exists o \in O(3), o(x_1) = x_2$), because the invariant representations and energy prediction are invariant under O(3) transformation. Therefore, injective mapping and maximum expressivity mean that function can differentiate inputs unequal in this sense.

**Encoding local environment.** Similar to that MPNN can encode neighbor nodes injectively on the graph, GNN-LF can encode neighbor nodes injectively in 3D space. Other models can also be analyzed from an encoding local environments perspective. GNNs for PES only collect messages from atoms within the sphere of radius $r_c$, so one message passing layer of them is equivalent to encoding the local environments in Definition 2.3. When mapping local environments *injectively*, a single message passing layer reaches maximum expressivity.

Some popular models are under-expressive. For example, as shown in Figure 2a, SchNet [4] only considers the distance between atoms and neglects the angular information, leading to the inability to differentiate some simple local environments. Moreover, Figure 2b illustrates that though DimeNet [5] adds angular information to message passing, its expressivity is still limited, which may be attributed to the loss of high-order geometric information like dihedral angle.

In contrast, no information loss will happen when we use the coordinates projected on the frame.

**Theorem 4.1.** *There exists a function $\chi$. Given a frame $\vec{E}_i$ of the atom $i$, $\chi$ encodes the local environment of atom $i$ injectively into atom $i$'s embeddings.*

$$\chi(\{(s_j, \vec{r}_{ij})|r_{ij} < r_c\}) = \rho(\sum_{r_{ij}<r_c} \psi(Concatenate(P_{\vec{E}_i}(\vec{r}_{ij}), s_j))). \qquad (4)$$

Theorem 4.1 shows that an ordinary message passing layer can encode local environments injectively with coordinate projection as an edge feature.

**Passing messages across local environments.** In physics, interaction between distant atoms is usually not negligible. Using one single message passing layer, which encodes atoms within cutoff radius only, leads to loss of such interaction. When using multiple message passing layers, GNN can pass messages between two distant atoms along a path of atoms and thus model the interaction.

However, passing messages in multiple steps may lead to loss of information. For example, in Figure 2c, two molecules are different as a part of the molecule rotates. However, the local environment will not change. So the node representations, the messages passed between nodes, and finally, the energy prediction will not change while two molecules have different energy. This problem will also happen in previous PES models [4, 5]. Loss of information in multi-step message passing is a fundamental and challenging problem even for ordinary GNN [13].

Nevertheless, the solution is simple in this special case. We can eliminate the information loss by *frame-frame projection*, i.e., projecting $\vec{E}_i$ (the frame of atom $j$) on $\vec{E}_j$ (the frame of atom $i$). For example, in Figure 2c, as the molecule rotates, frame vectors also rotate, leading to frame-frame projection change, so our model can differentiate them. We also prove the effectiveness of frame-frame projection in theory.

**Theorem 4.2.** *Let $\mathcal{G}$ denote the graph in which node $i$ represents the atom $i$ and edge $ij$ exists iff $r_{ij} < r_c$, where $r_c$ is the cutoff radius. Assuming frames exist, if $\mathcal{G}$ is a connected graph whose diameter is $L$, GNN with $L$ message passing layers as follows can encode the whole molecule $\{(s_j, \vec{r}_{ij})|j \in \{1, 2, ..., n\}\}$ injectively into the embedding of node $i$.*

$$\chi(\{(s_j, \vec{r}_{ij}, \vec{E}_j)|r_{ij} < r_c\}) = \rho(\sum_{r_{ij} < r_c} \psi(Concatenate(P_{\vec{E}_i}(\vec{r}_{ij}), P_{\vec{E}_i}(\vec{E}_j), s_j)). \tag{5}$$

Theorem 4.2 shows that an ordinary GNN can encode the whole molecule injectively with coordinate projection and frame-frame projection as edge features.

In conclusion, when frames exist, **even ordinary GNN can encode molecule injectively and thus reach maximum expressivity with coordinate projection and frame-frame projection.**

## 5 How to build a frame?

We propose frame generation method after discussing how to use frames because generation method's connection to expressivity is less direct. Whatever frame generation method is used, GNN-LF can keep expressivity as long as the frame does not degenerate. A frame degenerates iff it has less than three linearly independent vectors. This section provides one feasible frame generation method.

A straightforward idea is produce frames using invariant features of each atom, like the atomic number. However, function of invariant features must be invariant representations rather than equivariant frames. Therefore, we consider producing the frame from the local environment of each atom, which contains equivariant 3D coordinates. In Theorem 5.1, we prove that there exists a function mapping the local environment to an $O(3)$-equivariant frame.

**Theorem 5.1.** *There exists an $O(3)$-equivariant function $g$ mapping the local environment $LE_i$ to an equivariant representation in $\mathbb{R}^{3 \times 3}$. The output forms a frame if $\forall o \in O(3), o \neq I, o(LE_i) \neq LE_i$.*

Proof is in Appendix A.5. Frames produced by the function in Theorem 5.1 will not degenerate if local environments have no symmetry elements, like inversion centers, rotation axes, or mirror planes.

Building a frame for a symmetric local environment remains a problem in our current implementation but will **not seriously hamper our model**. Firstly, our model can produce reasonable output even with symmetric input and is provably more expressive than a widely used model SchNet [4] (see Appendix G). Secondly, symmetric molecules are rare and form a zero-measure set. In our two representative real-world datasets, less than $0.01\%$ of molecules (about ten molecules in the whole datasets of several hundred thousand molecules) are symmetric. Thirdly, symmetric geometry may be captured with a degenerate frame. As shown in Figure 3a, water is a symmetric molecule. We can use a frame with one vector to describe its geometry. Based on node identity features and relational pooling [28], we also propose a scheme in Appendix H to completely solve the expressivity loss caused by degeneration. However, for scalability, we do not use it in GNN-LF.

**A message passing layer for frame generation.** The existence of the frame generation function is proved in Theorem 4.2. Here we demonstrate how to implement it. There exists a universal framework for approximating $O(3)$-equivariant functions [15] which can be used to implement the function in Theorem 5.1. For scalability, we use a simplified form of that framework which has empirically good performance:

$$\vec{E}_i = \sum_{j \neq i, r_{ij} < r_c} g'(r_{ij}, s_j) \cdot \frac{\vec{r}_{ij}}{r_{ij}}, \tag{6}$$

where $g'$ maps invariant features and distance to invariant weights and the entire framework reduces to a message passing process. The derivation is detailed in Appendix B.

**Local frame vs global frame.** With the message passing framework in Equation 6, an individual frame, called *local frame*, is produced for each atom. These local frames can also be summed to

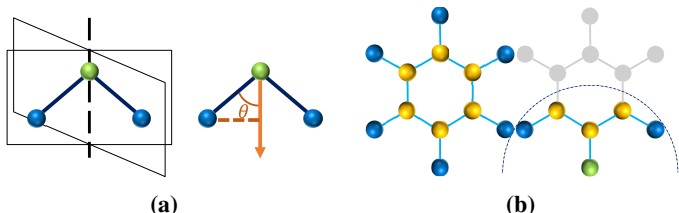

**(a)**                    **(b)**

**Figure 3:** (a) The left part shows the symmetry of the water molecule, which has a rotation axis. Its equivariant vectors must be parallel to the rotation axis. However, with a frame composed of only one vector, its geometry can be described. The right part shows that with the projection of $\vec{r}_{ij}$ on the frame and the distance between two atoms, the angle $\theta$ and the position of $j$ atom can be determined. (b) The left part is a molecule with central symmetry. Its global frame will be zero. However, when selected as the center (green), the atom's environment has no central symmetry.

produce a *global frame*.

$$\vec{E} = \sum_{i=1}^{n} \vec{E}_i. \tag{7}$$

The global frame can replace local frames and keep the invariance of energy prediction. All previous analysis will still be valid if the frame degeneration does not happen. However, the global frame is more likely to degenerate than local frames. As shown in Figure 3b, the benzene molecule has central symmetry and produces a zero global frame. However, when choosing each atom as the center, the central symmetry is broken, and a non-zero local frame can be produced. We further formalize this intuition and prove that the global frame is more likely to degenerate in Appendix I.

In conclusion, **we can generate local frames with a message passing layer**.

## 6  GNN with local frame

We formally introduce our GNN with local frame (GNN-LF) model. The whole architecture is detailed in Appendix C. The time and space complexity are $O(Nn)$, where $N$ is the number of atoms in the molecule, and $n$ is the maximum number of neighbor atoms of one atom.

**Notations.** Let $F$ denote the hidden dimension. We first convert the input features, coordinates $\vec{r} \in \mathbb{R}^{N \times 3}$ and atomic numbers $z \in \mathbb{N}^{N}$, to a graph. The initial node feature $s_i^{(0)} \in \mathbb{R}^{F}$ is an embedding of the atomic number $z_i$. Edge $ij$ has two features: the edge weight $w_{ij} = \text{cutoff}(r_{ij})$ (where cutoff means the cutoff function), and the radial basis expansion of the distance $d_{ij}^0 = \text{rbf}(r_{ij})$. Edge weight $w_{ij}$ is not necessary for expressivity. However, to ensure that the energy prediction is a smooth function of coordinates, messages passed among atoms must be scaled with $w_{ij}$ [19]. These special functions are detailed in Appendix C.

**Producing frame.** The message passing scheme for producing local frames implements Equation (6).

$$\vec{E}_i = \sum_{j \neq i, r_{ij} < r_c} w_{ij}(f_1(d_{ij}^0) \odot s_j) \cdot \frac{\vec{r}_{ij}}{r_{ij}}, \tag{8}$$

where $f_1$ is an MLP. Note that frame $\vec{E}_i \in \mathbb{R}^{F \times 3}$ in implementation is not restricted to have three vectors. The number of vectors equals the hidden dimension. The frame in $\mathbb{R}^{F \times 3}$ can be considered as an ensemble of $\frac{F}{3}$ frames in $\mathbb{R}^{3 \times 3}$, so this design will not hamper the expressivity.

**Coordinate projection** is as follows,

$$d_{ij}^1 = \frac{1}{r_{ij}} \vec{r}_{ij} \vec{E}_i^T. \tag{9}$$

The projection in implementation is scaled by $\frac{1}{r_{ij}}$ to decouple the distance information in $s_{ij}^{(e)}$.

**Frame-frame projection**. $\vec{E}_i \vec{E}_j^T$ is a large matrix. Therefore, we only use the diagonal elements of the projection. To keep the expressivity, we transform the frame with two ordinary linear layers.

$$d_{ij}^2 = \text{diag}(W_1 \vec{E}_j \vec{E}_i^T W_2^T). \tag{10}$$

Adding the projections to edge features, we get a graph with invariant features only.

**GNN working on the invariant graph**. The message passing scheme uses the form in Theorem 4.1. Let the $s_i^{(l)}$ denote the node representations produced by the $l^{\text{th}}$ message passing layers. $s_i^{(0)} = s_i$.

$$s_i^{(l)} = \rho\Big( \sum_{j \neq i, r_{ij} < r_c} w_{ij}(f_2(d_{ij}^0, d_{ij}^1, d_{ij}^2) \odot s_j^{(l-1)}) \Big), \tag{11}$$

where $\rho$ is an MLP. We further use a *filter decomposition* design as follows.

$$f_2(d_{ij}^0, d_{ij}^1, d_{ij}^2) = g_1(d_{ij}^0) \odot g_2(d_{ij}^1, d_{ij}^2). \tag{12}$$

The distance information $d_{ij}^0$ is easier to learn as it has been expanded with a set of bases, so a linear layer $g_1$ is enough. In contrast, projections need a more expressive MLP $g_2$.

**Sharing filters.** Generating different filters $f_2(d_{ij}^0, d_{ij}^1, d_{ij}^2)$ for each message passing layer is time-consuming. Therefore, we share filters between different layers. Experimental results show that sharing filters leads to minor performance loss and significant scalability gain.

**Gaps between the implementation and expressivity analysis.** Unlike the expressivity analysis in section 4, GNN-LF in implementation uses an ensemble of frames instead of only one, takes a different form of frame-frame projection, and does not constrain the generated frames to be orthogonal. In conclusion, experimental results show that GNN-LF with a single frame for each node can still achieve outstanding performance, and these implementation tricks are harmless to expressivity in theory. The discussion is detailed in Appendix M.

# 7 Experiment

In this section, we compare GNN-LF with existing models and do an ablation analysis. Experimental settings are detailed in Appendix D. The code is available at `https://github.com/GraphPKU/GNN-LF`. We report the mean absolute error (MAE) on the test set (the lower, the better). All our results are averaged over three random splits. Baselines' results are from their papers. The best and the second best results are shown in bold and underline respectively in tables.

## 7.1 Modeling PES

We first evaluate GNN-LF for modeling PES on the MD17 dataset [29], which consists of MD trajectories of small organic molecules. GNN-LF is compared with a manual descriptor model: FCHL [18] , invariant models: SchNet [4], DimeNet [5], GemNet [8], a model using irreducible representations: NequIP [9], and models using equivariant representations: PaiNN [7] and TorchMD [10]. The results are shown in Table 1. GNN-LF outperforms all the baselines on $9/16$ targets and achieves the second-best performance on other 6 targets. Our model leads to $10\%$ lower loss on average than GemNet, the best baseline. The outstanding performance verifies the effectiveness of the local frame method for modeling PES. Moreover, our model also uses **fewer parameters and only about** $30\%$ **time and** $10\%$ **GPU memory** compared with the baselines as shown in Appendix E.

## 7.2 Ablation study

We perform an ablation study to verify our model designs. The results are shown in Table 2.

On average, ablation of frame-frame projection (NoDir2) leads to $18\%$ higher MAE, which verifies the necessity of frame-frame projection. The column Global replaces the local frames with the global frame, resulting in $100\%$ higher loss, which verifies local frames' advantages over global frame. Ablation of filter decomposition (NoDecomp) leads to $10\%$ higher loss, indicating the advantage of separately processing distance and projections. Although using different filters for each message passing layer (NoShare) uses much more computation time ($1.67\times$) and parameters ($3.55\times$), it only leads to $0.01\%$ lower loss on average, illustrating that sharing filters does little harm to the expressivity.

## 7.3 Other chemical properties

Though designed for PES, our model can also predict other properties directly. The QM9 dataset [30] consists of 134k stable small organic molecules. The task is to use the atomic numbers and coordinates to predict properties of these molecules. We compare our model with invariant models:

**Table 1:** Results on the MD17 dataset. Units: energy ($\mathcal{E}$) (kcal/mol) and forces ($\mathcal{F}$) (kcal/mol/Å).

| Molecule | Target | FCHL | SchNet | DimeNet | GemNet | PaiNN | NequIP | TorchMD | **GNN-LF** |
|---|---|---|---|---|---|---|---|---|---|
| Aspirin | $\mathcal{E}$ | 0.182 | 0.37 | 0.204 | - | 0.167 | - | **0.124** | 0.1342 |
| | $\mathcal{F}$ | 0.478 | 1.35 | 0.499 | 0.2168 | 0.338 | 0.348 | 0.255 | **0.2018** |
| Benzene | $\mathcal{E}$ | - | 0.08 | 0.078 | - | - | - | **0.056** | 0.0686 |
| | $\mathcal{F}$ | - | 0.31 | 0.187 | **0.1453** | - | 0.187 | 0.201 | 0.1506 |
| Ethanol | $\mathcal{E}$ | 0.054 | 0.08 | 0.064 | - | 0.064 | - | 0.054 | **0.0520** |
| | $\mathcal{F}$ | 0.136 | 0.39 | 0.230 | 0.0853 | 0.224 | 0.208 | 0.116 | **0.0814** |
| Malonaldehyde | $\mathcal{E}$ | 0.081 | 0.13 | 0.104 | - | 0.091 | - | 0.079 | **0.0764** |
| | $\mathcal{F}$ | 0.245 | 0.66 | 0.383 | 0.1545 | 0.319 | 0.337 | 0.176 | **0.1259** |
| Naphthalene | $\mathcal{E}$ | 0.117 | 0.16 | 0.122 | - | 0.166 | - | **0.085** | 0.1136 |
| | $\mathcal{F}$ | 0.151 | 0.58 | 0.215 | 0.0553 | 0.077 | 0.097 | 0.060 | **0.0550** |
| Salicylic acid | $\mathcal{E}$ | 0.114 | 0.20 | 0.134 | - | 0.166 | - | **0.094** | 0.1081 |
| | $\mathcal{F}$ | 0.221 | 0.85 | 0.374 | 0.1048 | 0.195 | 0.238 | 0.135 | **0.1005** |
| Toluene | $\mathcal{E}$ | 0.098 | 0.12 | 0.102 | - | 0.095 | - | **0.074** | 0.0930 |
| | $\mathcal{F}$ | 0.203 | 0.57 | 0.216 | 0.0600 | 0.094 | 0.101 | 0.066 | **0.0543** |
| Uracil | $\mathcal{E}$ | 0.104 | 0.14 | 0.115 | - | 0.106 | - | **0.096** | 0.1037 |
| | $\mathcal{F}$ | 0.105 | 0.56 | 0.301 | 0.0969 | 0.139 | 0.173 | 0.094 | **0.0751** |
| average | rank | 3.93 | 6.63 | 5.38 | 2.00 | 4.36 | 5.25 | 2.25 | 1.75 |

**Table 2:** Ablation results on the MD17 dataset. Units: energy ($\mathcal{E}$) (kcal/mol) and forces ($\mathcal{F}$) (kcal/mol/Å).

| Molecule | Target | GNN-LF | NoDir2 | Global | NoDecomp | GNN-LF | Noshare |
|---|---|---|---|---|---|---|---|
| Aspirin | $\mathcal{E}$ | **0.1342** | 0.1499 | 0.2280 | 0.1411 | **0.1342** | 0.1364 |
| | $\mathcal{F}$ | **0.2018** | 0.2909 | 0.6894 | 0.2622 | 0.2018 | **0.1979** |
| Benzene | $\mathcal{E}$ | **0.0686** | 0.0716 | 0.0972 | 0.0688 | **0.0686** | 0.0713 |
| | $\mathcal{F}$ | 0.1506 | 0.1583 | 0.3520 | **0.1499** | 0.1506 | 0.1507 |
| Ethanol | $\mathcal{E}$ | 0.0520 | 0.0532 | 0.0556 | **0.0518** | 0.0533 | **0.0514** |
| | $\mathcal{F}$ | **0.0814** | 0.1056 | 0.1465 | 0.0895 | 0.0814 | **0.0751** |
| Malonaldehyde | $\mathcal{E}$ | **0.0764** | 0.0776 | 0.0923 | 0.0786 | **0.0764** | 0.0790 |
| | $\mathcal{F}$ | **0.1259** | 0.1728 | 0.3194 | 0.1422 | 0.1259 | **0.1210** |
| Naphthalene | $\mathcal{E}$ | **0.1136** | 0.1152 | 0.1276 | 0.1171 | **0.1136** | 0.1168 |
| | $\mathcal{F}$ | **0.0550** | 0.0834 | 0.2069 | 0.0683 | 0.0550 | **0.0547** |
| Salicylic acid | $\mathcal{E}$ | **0.1081** | 0.1087 | 0.1224 | 0.1123 | **0.1081** | 0.1091 |
| | $\mathcal{F}$ | **0.1048** | 0.1412 | 0.2890 | 0.1399 | 0.1048 | **0.1012** |
| Toluene | $\mathcal{E}$ | **0.0930** | 0.0942 | 0.1000 | 0.0932 | **0.0930** | 0.0942 |
| | $\mathcal{F}$ | **0.0543** | 0.0770 | 0.1659 | 0.0695 | 0.0543 | **0.0519** |
| Uracil | $\mathcal{E}$ | **0.1037** | 0.1069 | 0.1075 | 0.1053 | **0.1037** | 0.1042 |
| | $\mathcal{F}$ | **0.0751** | 0.0964 | 0.1901 | 0.0825 | **0.0751** | 0.0754 |

SchNet [4], DimeNet++ [31], ComENet [32], a model using irreducible representations: Cormorant [6], SE(3)-Transformer [22], and models using equivariant representations: EGNN [33], PaiNN [7] and TorchMD [10]. Results are shown in Table 3. Our model outperforms all other models on 7/12 tasks and achieves the second-best performance on 4/5 left tasks, which illustrates that the local frame method has the potential to be applied to other fields.

## 8 Conclusion

This paper proposes GNN-LF, a simple and effective molecular potential energy surface model. It introduces a novel local frame method to decouple the symmetry requirement and capture rich geometric information. In theory, we prove that even ordinary GNNs can reach maximum expressivity with the local frame method. Furthermore, we propose ways to construct local frames. In experiments,

**Table 3:** Results on the QM9 dataset. SE(3)-T is short for SE(3)-Transformer

| Target | Unit | SchNet | DimeNet++ | ComENet | Cormorant | SE(3)-T | PaiNN | EGNN | Torchmd | **GNN-LF** |
|--------|------|--------|-----------|---------|-----------|---------|-------|------|---------|------------|
| $\mu$ | D | 0.033 | 0.0297 | 0.0245 | 0.038 | 0.051 | 0.012 | 0.029 | **0.002** | 0.013 |
| $\alpha$ | $a_0^3$ | 0.235 | 0.0435 | 0.0452 | 0.085 | 0.142 | 0.045 | 0.071 | **0.01** | 0.0426 |
| $\epsilon_{\text{HOMO}}$ | meV | 41 | 24.6 | 23.1 | 34 | 35 | 27.6 | 29 | **21.2** | 23.5 |
| $\epsilon_{\text{LUMO}}$ | meV | 34 | 19.5 | 19.8 | 38 | 33 | 20.4 | 25 | 17.8 | **17.0** |
| $\Delta\epsilon$ | meV | 63 | **32.6** | **32.4** | 61 | 53 | 45.7 | 48 | 38 | 37.1 |
| $\langle R^2 \rangle$ | $a_0^2$ | 0.073 | 0.331 | 0.259 | 0.961 | - | 0.066 | 0.106 | **0.015** | 0.037 |
| ZPVE | meV | 1.7 | 1.21 | 1.2 | 2.027 | - | 1.28 | 1.55 | 2.12 | **1.19** |
| $U_0$ | meV | 14 | 6.32 | 6.59 | 22 | - | 5.85 | 11 | 6.24 | **5.30** |
| $U$ | meV | 19 | 6.28 | 6.82 | 21 | - | 5.83 | 12 | 6.3 | **5.24** |
| $H$ | meV | 14 | 6.53 | 6.86 | 21 | - | 5.98 | 12 | 6.48 | **5.48** |
| $G$ | meV | 14 | 7.56 | 7.98 | 20 | - | 7.35 | 12 | 7.64 | **6.84** |
| $C_v$ | cal/mol/K | 0.033 | 0.023 | 0.024 | 0.026 | 0.054 | 0.024 | 0.031 | 0.026 | **0.022** |

our model outperforms all baselines in both scalability (using only $30\%$ time and $10\%$ GPU memory) and accuracy ($10\%$ lower loss). Ablation study also verifies the effectiveness of our designs.

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

## A   Proofs

Due to repulsive force, atoms cannot be too close to each other in stable molecules. Therefore, we assume that there exist an upper bound $N$ of the number of neighbor atoms.

### A.1   Proof of Lemma 2.1

*Proof.* For all function $g$, invariant representation $s$, transformation $o \in O(3)$, $g(s(z, \vec{r}o^T)) = g(s)$. Therefore, $g(s)$ is an invariant representation.

For all invariant representation $s$, equivariant representation $\vec{v}$, and transformation $o \in O(3)$,

$$s(z, \vec{r}o^T) \cdot \vec{v}(z, \vec{r}o^T) = s(z, \vec{r}) \cdot (\vec{v}(z, \vec{r})o^T) = (s(z, \vec{r}) \cdot \vec{v}(z, \vec{r}))o^T. \tag{13}$$

Therefore, $s \cdot \vec{v}$ is an equivariant representation.

For all equivariant representations $\vec{v}$,

$$\vec{v}(z, \vec{r}o^T)\vec{E}(z, \vec{r}o^T)^T = \vec{v}(z, \vec{r})o^T o\vec{E}(z, \vec{r})^T = \vec{v}(z, \vec{r})\vec{E}(z, \vec{r})^T \tag{14}$$

$P_{\vec{E}}$ is inversible because

$$P_{\vec{E}}(\vec{v})\vec{E} = \vec{v}\vec{E}^T\vec{E} = \vec{v}. \tag{15}$$

For all invariant representations $s \in \mathbb{R}^{F \times 3}$,

$$s(z, \vec{r}o^T)\vec{E}(z, \vec{r}o^T) = s(z, \vec{r})\vec{E}(z, \vec{r})o^T. \tag{16}$$

Similarly,

$$\vec{v}(z, \vec{r}o^T)\vec{v}'(z, \vec{r}o^T)^T = \vec{v}(z, \vec{r})o^T o\vec{v}'(z, \vec{r})^T = \vec{v}(z, \vec{r})\vec{v}'(z, \vec{r})^T. \tag{17}$$

Therefore, projection on general equivariant representations can also produce invariant representation.
□

### A.2   Proof of Proposition 4.1

*Proof.* Assume that $s$ is an invariant representation.

$$\tilde{g}(s) = P_{\vec{E}}(g(P_{\vec{E}}^{-1}(s))) \tag{18}$$

$$= g(s(\vec{E}(z, \vec{r})^{-1})^T))\vec{E}(z, \vec{r})^T. \tag{19}$$

The representation $\tilde{g}(s)$ can be written as a function of $(z, \vec{r})$. Then, we have

$$\forall o \in O(3), \tilde{g}(s)(z, \vec{r}o^T) = g(s(\vec{E}(z, \vec{r}o^T)^{-1})^T))\vec{E}(z, \vec{r}o^T)^T \tag{20}$$

$$= g(s(\vec{E}(z, \vec{r})^{-1})^T)o^T)o\vec{E}(z, \vec{r})^T \tag{21}$$

$$= g(s(\vec{E}(z, \vec{r})^{-1})^T))o^T o\vec{E}(z, \vec{r})^T \tag{22}$$

$$= g(s(\vec{E}(z, \vec{r})^{-1})^T))\vec{E}(z, \vec{r})^T \tag{23}$$

$$= \tilde{g}(s)(z, \vec{r}). \tag{24}$$

Therefore, $\tilde{g}(s)$ is also an invariant representation.
□

## A.3 Proof of Theorem 4.1

We first prove that when the multiset of invariant features and coordinate projections equal, the multiset of invariant features and coordinates are just distinguished from each other with an orthogonal transformation.

**Lemma A.1.** *Given two frames $\vec{E}_1$, $\vec{E}_2$, and two sets of atoms $\{(s_{1,i}, \vec{r}_{1,i} | i = 1, 2, ..., n\}$, $\{(s_{2,i}, \vec{r}_{2,i} | i = 1, 2, ..., n\}$. If $\{(s_{1,i}, P_{\vec{E}_1}(\vec{r}_{1,i}) | i = 1, 2, ..., n\} = \{(s_{2,i}, P_{\vec{E}_2}(\vec{r}_{2,i}) | i = 1, 2, ..., n\}$, there exists $o \in O(3)$, $\{(s_{1,i}, \vec{r}_{1,i} | i = 1, 2, ..., n\} = \{(s_{2,i}, \vec{r}_{2,i} o^T | i = 1, 2, ..., n\}$*

*Proof.* As $\{(s_{1,i}, P_{\vec{E}_1}(\vec{r}_{1,i}) | i = 1, 2, ..., n\} = \{(s_{2,i}, P_{\vec{E}_2}(\vec{r}_{2,i}) | i = 1, 2, ..., n\}$, there exists permutation $\pi : \{1, 2, ..., n\} \to \{1, 2, ..., n\}$, so that

$$s_{1,i} = s_{1,\pi(i)}, P_{\vec{E}_1}(\vec{r}_{1,i}) = P_{\vec{E}_2}(\vec{r}_{2,\pi(i)}) \tag{25}$$

$$s_{1,i} = s_{1,\pi(i)}, \vec{r}_{1,i}\vec{E}_1^T = \vec{r}_{2,\pi(i)}\vec{E}_2^T \tag{26}$$

$$s_{1,i} = s_{1,\pi(i)}, \vec{r}_{1,i} = \vec{r}_{2,\pi(i)}\vec{E}_2^T\vec{E}_1. \tag{27}$$

As $\vec{E}_1, \vec{E}_2$ are both orthogonal matrix, $\vec{E}_2^T\vec{E}_1 \in O(3)$. Let $o$ denotes $\vec{E}_2^T\vec{E}_1$,

$$\{(s_{1,i}, \vec{r}_{1,i} | i = 1, 2, ..., n\} = \{(s_{2,i}, \vec{r}_{2,i}o^T | i = 1, 2, ..., n\}. \tag{28}$$

$\square$

According to [34], there exists $\rho$ and $\psi$ so that

$$\rho(\sum_{r_{ij} < r_c} \psi(\text{Concatenate}(P_{\vec{E}_i}(\vec{r}_{ij}), s_j))) \tag{29}$$

encoding $\{(P_{\vec{E}_i}(\vec{r}_{ij}), s_j) | r_{ij} < r_c\}$ injectively. Let $\chi$ denote this function. According to Lemma A.1, $\chi$ encodes local environments injectively when the difference caused by orthogonal transformation is neglected.

## A.4 Proof of Theorem 4.2

Notation: Given a molecule with atom coordinates $\vec{r} \in \mathbb{R}^{N \times 3}$ and atomic features (like embedding of atomic number) $s \in \mathbb{R}^{N \times F}$, let $\mathcal{G}$ denote the undirected graph corresponding to the molecule. Node $i$ in $\mathcal{G}$ represents the atom $i$ in the molecule. $\mathcal{G}$ has edge $ij$ iff $r_{ij} < r_c$, where $r_c$ is the cutoff radius. Let $d(\mathcal{G}, i, j)$ denote the shortest path distance between node $i$ and $j$ in graph $\mathcal{G}$.

Note that a single layer defined in Equation 5 can still encode the local environment, as extra frame-frame projection cannot lower the expressivity.

**Lemma A.2.** *Given a frame $\vec{E}$, with suitable functions $\rho$ and $\psi$, $\chi$ defined in Equation 5 encodes the local environment injectively.*

*Proof.* According to Theorem 4.1, there exists $\rho'$ and $\psi'$ so that

$$\rho(\sum_{r_{ij} < r_c} \psi(\text{Concatenate}(P_{\vec{E}_i}(\vec{r}_{ij}), P_{\vec{E}_i}(\vec{E}_j), s_j)) \tag{30}$$

can encode local environment injectively. Let $\rho$ equals to $\rho'$,

$$\psi(\text{cat}(P_{\vec{E}_i}(\vec{r}_{ij}), P_{\vec{E}_i}(\vec{E}_j), s_j)) = \psi'(\text{cat}(P_{\vec{E}_i}(\vec{r}_{ij}), s_j)). \tag{31}$$

Then,

$$\chi(\{(s_j, \vec{r}_{ij}, \vec{E}_j) | r_{ij} < r_c\}) = \rho'(\sum_{r_{ij} < r_c} \psi'(\text{cat}(P_{\vec{E}_i}(\vec{r}_{ij}), s_j)) \tag{32}$$

encodes local environment injectively. $\square$

Now we begin to prove the Theorem 4.2.

*Proof.* We use cat to represent concatenate throughout the proof. Let $N(i)_l$ denote $\{j|d(\mathcal{G}, i, j) \leq l\}$. The $l^{\text{th}}$ message passing layer has the following form.

$$s_i^{(l)} = \rho_l \left( \sum_{j \in N_1(i)} \psi_l(\text{cat}(P_{\vec{E}_i}(\vec{r}_{ij}), P_{\vec{E}_i}(\vec{E}_j), s_j^{(l-1)})) \right), \tag{33}$$

where $s_i^{(0)} = s_i$.

By enumeration on $l$, we prove that there exist $\rho_l, \psi_l$ so that $s_i^{(l)} = \varphi(\{\text{cat}(s_j, P_{\vec{E}_i}(\vec{r}_{ij}))|j \in N_l(i)\})$.

We first define some auxiliary functions.

According to [34], there exists a multiset function $\varphi$ mapping a multiset of invariant representations to an invariant representation injectively. $\varphi$ can have the following form

$$\varphi(\{x_i|i \in \mathbb{I}\}) = \sum_i \psi(x_i), \tag{34}$$

where $\mathbb{I}$ is some finite index set. As $\varphi$ is injective, it has an inverse function.

We define function $m, m', m''$ to extract invariant representation from concatenated node features.

$$m(\text{cat}(P_{\vec{E}_i}(\vec{r}_{ij}), P_{\vec{E}_i}(\vec{E}_j), s_j^{(0)})) = \text{cat}(s_j^{(0)}, P_{\vec{E}_i}(\vec{r}_{ij})). \tag{35}$$

$$m'(\text{cat}(s_j, P_{\vec{E}_i}(\vec{r}_{ij}))) = s_j. \tag{36}$$

$$m''(\text{cat}(s_j, P_{\vec{E}_i}(\vec{r}_{ij}))) = P_{\vec{E}_i}(\vec{r}_{ij}). \tag{37}$$

Last but not least, there exist a function $T$ transforming coordinate projections from one frame to another frame.

$$T(P_{\vec{E}_i}(\vec{r}_{ij}), P_{\vec{E}_i}(\vec{E}_j), P_{\vec{E}_j}(\vec{r}_{jk})) = P_{\vec{E}_i}(\vec{r}_{ij}) + P_{\vec{E}_j}(\vec{r}_{jk})P_{\vec{E}_i}(\vec{E}_j) = P_{\vec{E}_i}(\vec{r}_{ik}) \tag{38}$$

$l = 1$: let $\psi_1 = \psi \cdot m$ , $\rho_1$ is identity mapping.

$l > 1$: Assume for all $l' < l$, $s_i^{(l')} = \varphi(\{\text{cat}(s_j, P_{\vec{E}_i}(\vec{r}_{ij}))|j \in N_{l'}(i)\})$.

$\psi_l$ has the following form.

$$\psi_l(\text{cat}(P_{\vec{E}_i}(\vec{r}_{ij}), P_{\vec{E}_i}(\vec{E}_j), s_j^{(l-1)})) = \psi(\varphi($$
$$\{\text{cat}(m'(x), T(P_{\vec{E}_i}(\vec{r}_{ij}), P_{\vec{E}_i}(\vec{E}_j), m''(x))|x \in \varphi^{-1}(s_j^{(l-1)})\})). \tag{39}$$

Therefore

$$\psi_l(\text{cat}(P_{\vec{E}_i}(\vec{r}_{ij}), P_{\vec{E}_i}(\vec{E}_j), s_j^{(l-1)})) = \psi(\varphi(\{\text{cat}(s_k, P_{\vec{E}_i}(\vec{r}_{ik}))|k \in N_{l-1}(j)\})). \tag{40}$$

Note $\psi_l$ transforms coordinate projection from an old frame to a new frame.

Therefore, the input of $\rho_l$, namely $a_i^{(l)}$, has the following form.

$$a_i^{(l)} = \sum_{j \in N(i)} \psi_l(\text{cat}(P_{\vec{E}_i}(\vec{r}_{ij}), P_{\vec{E}_i}(\vec{E}_j), s_j^{(l-1)})) \tag{41}$$

$$= \sum_{j \in N(i)} \psi(\varphi(\{\text{cat}(s_k, P_{\vec{E}_i}(\vec{r}_{ik}))|k \in N_{l-1}(j)\})) \tag{42}$$

$$= \varphi(\{\varphi(\{\text{cat}(s_k, P_{\vec{E}_i}(\vec{r}_{ik}))|k \in N_{l-1}(j)\})|j \in N(i)\}) \tag{43}$$

We can transform $a_i^{(l)}$ to a set of set of invariant representations with the following function.

$$\eta(a_i^{(l)}) = \{\varphi^{-1}(s)|s \in \varphi^{-1}(a_i^{(l)})\}. \tag{44}$$

Therefore, $\eta(a_i^{(l)}) = \{\{\text{cat}(s_k, P_{\vec{E}_i}(\vec{r}_{ik}))|k \in N_{l-1}(j)\}|j \in N(i)\}$

We can use another function $\iota$ unions invariant representation sets in set $\mathbb{S}$ to a set of invariant representation.

$$\iota(\mathbb{S}) = \bigcup_{s \in \mathbb{S}} s. \tag{45}$$

$\rho_l$ has the following form.

$$\rho_l(a_i^{(l)}) = \varphi \circ \iota \circ \eta(a_i^{(l)}). \tag{46}$$

Therefore, the output is

$$\rho_l(a_i^{(l)}) = \varphi(\{\text{cat}(s_k, P_{\vec{E}_i}(\vec{r}_{ik})) | k \in N_l(i)\}\}) \tag{47}$$

Therefore, $\forall l \in \mathbb{N}$, there exists $\rho_l, \psi_l$ so that $s_i^{(l)} = \varphi(\{\text{cat}(s_j, P_{\vec{E}_i}(\vec{r}_{ij})) | j \in N_l(i)\})$.

As $L$ is the diameter of $\mathcal{G}$, $s_i^{(L)} = s_i^{(l)} = \varphi(\{\text{cat}(s_j, P_{\vec{E}_i}(\vec{r}_{ij})) | j \in N_L(i)\}) = \varphi(\{\text{cat}(s_j, P_{\vec{E}_i}(\vec{r}_{ij})) | j \in \{1, 2, ..., n\}\})$. As $\varphi$ is an injective function, GNN with $L$ message passing layers defined in Equation 5 can encode the $\{(s_i, P_{\vec{E}_i}\vec{r}_{ij}) | j \in \{1, 2, ..., n\}\}$ injectively to $s_i^{(L)}$. According to Lemma A.1, this GNN encodes the whole molecule $\{(s_i, \vec{r}_{ij}) | j \in \{1, 2, ..., n\}\}$ when the difference caused by orthogonal transformation is neglected. $\qquad \square$

### A.5  Proof of Theorem 5.1

*Proof.* (1) We first prove there exists an O(3)-equivariant function $g$ mapping the local environment $LE_i$ to a frame $\vec{E}_i \in \mathbb{R}^{3 \times 3}$. The frame has full rank if there does not exist $o \in O(3), o \neq I, o(LE_i) = LE_i$.

Let $\gamma$ denote a function mapping local environments to sets of vectors.

$$\gamma(\{(\vec{r}_{ij}, s_j) | r_{ij} < r_c\}) = \{\text{Concatenate}(s_j \vec{r}_{ij}, \vec{r}_{ij}) | r_{ij} < r_c\}, \tag{48}$$

in which $s_j$ is reshaped as $F \times 1$, $\vec{r}_{ij}$ is of shape $1 \times 3$. $\gamma$ is O(3)-equivariant. Therefore, we discuss the aggregation function on a set of equivariant representation, denoted as $\{\vec{u}_i | i = 1, 2, ..., n, \vec{u}_i \in \mathbb{R}^{F \times 3}\}$.

Assume that $V = \{\{\vec{u}_i | i = 1, 2, ..., n, \vec{u}_i \in \mathbb{R}^{F \times 3}\} | n = 1, 2, ..., N\}$, where $N$ is the upper bound of the size of local environment, is the set of sets of equivariant messages in local environment.

An equivalence relation can be defined on $V$: $v_1 \in V, v_2 \in V, v_1 \sim v_2$ iff there exists $o \in O(3), o(v_1) = v_2$. Let $\tilde{V} = V/\sim$ denote the quotient set. For each equivalence class $[v]$ with no symmetry, a representative $v$ can be selected. We can define a function $r : \tilde{V} - \{[v] | [v] \in \tilde{V}, \exists v \in [v], o \in O(3), o \neq I, o(v) = v\} \to V$ as $r([v]) = v$ mapping each equivalence class with no symmetry to its representative. For a message set with no symmetry, the transformation from its representative to it is also unique. Let $h : V - \{v | v \in V, \exists o \in O(3), o \neq I, o(v) = v\} \to O(3)$. $h(v) = o, o(r([v])) = v$.

Therefore, the function $g$ can take the form as follows.

$$g(v) = \begin{cases} \begin{bmatrix} 0 & 0 & 0 \\ 0 & 0 & 0 \\ 0 & 0 & 0 \end{bmatrix} & \text{if there exists } o \in O(3), o \neq I, ov = v \\ h(v)^T & \text{otherwise} \end{cases}$$

Therefore, $g \circ \gamma$ is the required function.

We further detail how to select the representative elements: We first define a linear order relation $\leq_l$ in $V$. If $v_1, v_2 \in V, |v_1| < |v_2|, v_1 <_l v_2$. So we only consider the order relation between two sets of the same size $n$.

We first define a function $\varphi$ mapping message set to a sequence injectively.

$$\varphi(\{u_i | i = 1, 2, ..., n, u_i \in \mathbb{R}^{F \times 3}\}) = [\text{flatten}(u_{\pi(i)}) | i = 1, 2, ..., n,$$
$$\pi \text{ is a permutation sorting } u_i \text{ by lexicographical order}]. \tag{49}$$

Forall $v_1, v_2 \in V, |v_1| = |v_2|, v_1 \leq_l v_2$ iff $\varphi(v_1) \leq \varphi(v_2)$ by lexicographical order. As the size of local environent is bounded, the sequence is also of a finite length. Therefore, the lexicographical order and thus the linear order relation $\leq_l$ are well-defined.

All permutations of $\{1, 2, ..., n\}$ form a permutation set $\Pi_n$.

For all $[v] \in \tilde{V}$, let $r([v]) = \arg\min_{v' \in [v]} \varphi(v')$. To illustrate the existence of such minimal sequence, we reform it.

$$min_{v' \in [v]} \varphi(v') = \min_{\pi \in \Pi_n, o \in O(3)} S(o, \pi) \tag{50}$$

$$= \min_{\pi \in \Pi_n} \min_{o \in O(3)} S(o, \pi), \tag{51}$$

where $S(o, \pi) = [\text{flatten}(u_{\pi(i)} o^T)|i = 1, 2, ..., n]$. Each element of this sequence is continuous to $o$.

We first fix the $\pi$. As O(3) is a compact group, $\arg\min_{o \in O(3)} S(o, \pi)_1$ exists. Let $L_1 = \{o|o \in O(3), S(o, \pi)_1 = \min_{o' \in O(3)} S(o', \pi)_1\}$ is still a compact set. Therefore, $\arg\min_{o \in L_1} S(o, \pi)_1$ exists. Let $L_2 = \{o|o \in L_1, S(o, \pi)_2 = \min_{o' \in L_1} S(o', \pi)_2\}$. Similarly, $L_3, L_4, ..., L_{3Fn}$ can be defined and they are non-empty set. Forall $o_1, o_2 \in L_{3Fn}$, as $S(o_1, \pi) \leq S(o_2, \pi)$ and $S(o_2, \pi) \leq S(o_1, \pi)$ by lexicographical order, $S(o_2, \pi) = S(o_1, \pi)$ and thus $o_1(v) = o_2(v)$. If v has no symmetry, $\forall o \in O(3), o \neq I, o(v) \neq v, o_1(v) = o_2(v) \implies o_1 = o_2$. Therefore, $L_{3Fn}$ contains a unique element $o_v^{(0)}$ and $\min_{o \in O(3)} S(o, \pi)$ is unique.

As $\Pi_n$ is a finite set, if $\min_{o \in O(3)} S(o, \pi)$ exist for all $\pi \in \Pi_n$, $\min_{\pi \in \Pi_n} \min_{o \in O(3)} S(o, \pi)$ must exist. Therefore the minimal sequence exists. As $\leq_l$ is a linear order, the minimal sequence is unique. With the unique sequence, the unique representative can be determined.

(2) Then we prove there does not exist $o \in O(3), o \neq I, o(LE_i) = LE_i$ if the frame has full rank.

The frame $\vec{E}$ is a function of local environment. If there exists

$$o \in O(3), o(LE_i) = LE_i.$$

Then $\vec{E}(o(LE_i)) = \vec{E}(LE_i)o^T = \vec{E}(LE_i)$.

As $\vec{E}$ is an invertible matrix, $o = I$. Therefore, $LE_i$ has no symmetry.

$\square$

## B  Derivation of the message passing section for frame

The framework proposed by Villar et al. [15] is

$$h_n(\{\vec{m}_{i1}, \vec{m}_{i2}, \vec{m}_{i2}, ..., \vec{m}_{in}\}) = \sum_{j=1}^{n} g(\vec{m}_{ij}, \{\vec{m}_{i1}, ..., \vec{m}_{in}\} - \{\vec{m}_{ij}\}) \cdot \vec{m}_{ij}, \tag{52}$$

where $h_n$ is the aggregation function for $n$ messages. $g$ is a O(3)-invariant functions. We can further reform it.

$$h_n(\{\{\vec{m}_{i1}, \vec{m}_{i2}, ..., \vec{m}_{in}\}\}) = \sum_{j=1}^{n} g_1^{(n)}(g_2^{(n)}(\vec{m}_{ij}), h_{n-1}(\{\vec{m}_{i1}, \vec{m}_{i2}, ..., \vec{m}_{i,n}\} - \{\vec{m}_{ij}\}))\vec{m}_{ij}, \tag{53}$$

where $g_1^{(n)}, g_2^{(n-1)}$ are two O(3)-invariant functions. With this equation, we can recursively build $n$ message aggregation function $h_n$ with $h_{n-1}$. Its universal approximation power has been proved in [15].

However, as they can have varied numbers of neighbors, different nodes have to use different aggregation functions, which is hard to implement. Therefore, we desert the recursive term $h_{n-1}$.

$$h_n(\{\{\vec{m}_{i1}, \vec{m}_{i2}, ..., \vec{m}_{in}\}\}) = \sum_{j=1}^{n} g(\vec{m}_{ij})\vec{m}_{ij}. \tag{54}$$

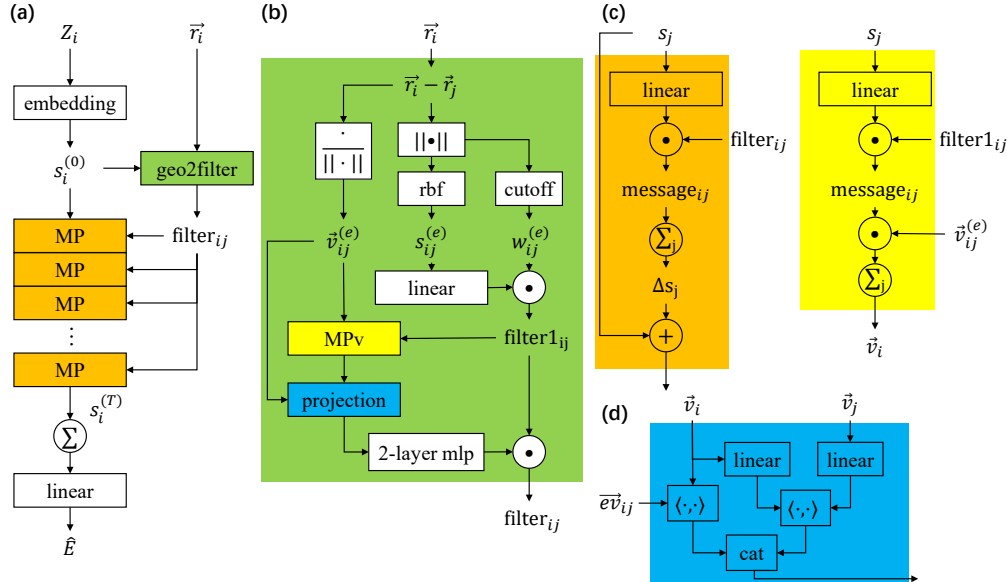

**Figure 4:** The architecture of GNN-LF. (a) The full architecture of GNN-LF contains four parts: an embedding block, a geo2filter block, message passing (MP) layers, and an output module. Embedding block consists of an embedding layer converting atomic numbers to learnable tensors and a neighborhood embedding block proposed by Thölke and Fabritiis [10]. (b) The geo2filter block builds a graph with the coordinates of atoms, passes messages to produce local frames, projects equivariant features onto the frames, and uses edge invariant features to produce edge filters. (c) A message passing layer filters atom representations with edge filters to produce messages and aggregates these messages to update atom embeddings. (d) The projection block produces $d^1, d^2$ and concatenates them.

The message $\vec{m}_{ij}$ can have the form Concatenate$(1, s) \cdot \vec{r}_{ij}$ in Theorem 4.1. As $g$ is an invariant function, we can further simplify Equation 54.

$$h_n(\{\{\vec{m}_{i1}, \vec{m}_{i2}, ..., \vec{m}_{in}\}\}) = \sum_{j=1}^{n} g'(r_{ij}, s_j) \frac{\vec{r}_{ij}}{r_{ij}}, \tag{55}$$

where $g'$ is a function mapping invariant representations to invariant representations.

## C Archtecture of GNN-LF

The full architecture is shown in Figure 4.

Following Thölke and Fabritiis [10], we also use a neighborhood embedding block which aggregates neighborhood information as the initial atom feature.

$$s_i^{(0)} = \text{Emb}_1(z_i) + \sum_{r_{ij} < r_c} Emb_2(z_j) \odot f(d_{ij}^0). \tag{56}$$

where $\text{Emb}_1$ and $\text{Emb}_2$ are two embedding layers and $f$ is the filter function.

These special functions are proposed by previous methods [19, 35].

$$\text{cutoff}(r) = \begin{cases} \frac{1}{2}(1 + \cos \frac{\pi r}{r_c}), r < r_c \\ 0, r > r_c \end{cases} \tag{57}$$

$$\text{rbf}_k(r_{ij}) = e^{-\beta_k(\exp(-r_{ij}) - \mu_k)^2}, \tag{58}$$

where $\beta_k, \mu_k$ are coefficients of the $k^{\text{th}}$ basis.

For PES tasks, the output module is a sum pooling and a linear layer. Other invariant prediction tasks can also use this module. However, on the QM9 dataset, we design special output modules for two properties. For dipole moment $\mu$, given node representations $[s_i | i = 1, 2, ..., N]$ and atom coordinates $[\vec{r}_i | i = 1, 2, ..., N]$, our prediction is as follows.

$$\hat{\mu} = |\sum_i (q_i - \text{average}_j(q_j))\vec{r}_i|, \tag{59}$$

where $q_i \in \mathbb{R}$, the prediction of charge, is function of $s_i$. We use a linear layer to convert $s_i$ to $q_i$. As the whole molecule is electroneutral, we use $q_i - \text{average}_j(q_j)$.

For electronic spatial extent $\langle R^2 \rangle$, we make use of atom mass (known constants) $[m_i | i = 1, 2, ..., N]$. The output module is as follows.

$$\vec{r}_c = \frac{\sum_i m_i \vec{r}_i}{\sum_i m_i} \tag{60}$$

$$\langle \hat{R^2} \rangle = |\sum_i x_i |\vec{r}_i - \vec{r}_c|^2|, \tag{61}$$

where $x_i \in \mathbb{R}$ is an invariant representation feature of node $i$. We also use a linear layer to convert $s_i$ to $x_i$.

**Table 4:** The training, inference time and the GPU memory consumption of random batches of 32 molecules (16 molecules for GemNet) from the MD17 dataset. The format is training time in ms/inference time in ms/inference GPU memory comsumption in MB. $N$ denotes the number of atoms in the molecule, and $n$ means an atom's maximum number of neighbors.

|  | DimeNet | GemNet | torchmd | GNN-LF | NoShare |
|---|---|---|---|---|---|
| number of parameters | 2.1 M | 2.2 M | 1.3 M | 0.8M~1.3M | 2.4M~5.3M |
| time complexity | $O(Nn^2)$ | $O(Nn^3)$ | $O(Nn)$ | $O(Nn)$ | $O(Nn)$ |
| aspirin | 727/133/5790 | 2823/612/15980 | 188/32/2065 | 65/10/279 | 142/22/883 |
| benzene | 669/ 94/1831 | 2242/393/3761 | 478/33/918 | 29/ 8/95 | 40/11/213 |
| ethanol | 672/ 95/784 | 2256/344/1565 | 417/32/532 | 59/ 8/54 | 76/11/115 |
| malonaldehyde | 657/ 88/784 | 2237/355/1565 | 753/32/532 | 57/ 7/68 | 68/10/127 |
| naphthalene | 614/112/4470 | 2613/498/11661 | 265/32/1694 | 61/ 9/175 | 97/15/491 |
| salicylic_acid | 619/ 92/3489 | 2577/430/8182 | 239/34/1418 | 59/ 9/176 | 79/15/424 |
| toluene | 595/113/3148 | 2495/423/7153 | 896/45/1322 | 62/ 8/176 | 83/15/458 |
| uracil | 595/107/1782 | 2165/354/3735 | 118/32/907 | 66/ 8/99 | 87/14/302 |
| average | 643/104/2760 | 2426/426/6700 | 419/34/1174 | 57/ 9/140 | 84/14/377 |

**Table 5:** The inference time and the GPU memory consumption of random batches of 32 molecules from the QM9 dataset and $U_0$ target. The format is inference time in ms/inference GPU memory comsumption in MB.

|  | EGNN | ComENet | GNN-LF |
|---|---|---|---|
| number of parameters | 0.75M | 4.2M | 1.7M |
| GPU memory | 1105 | 356 | 329 |
| inference time (ms) | 4.2 | 11.9 | 6.6 |

# D   Experiment settings

**Computing infrastructure.** We leverage Pytorch for model development. Hyperparameter searching and model training are conducted on an Nvidia A100 GPU. Inference times are calculated on an Nvidia RTX 3090 GPU.

**Training process.** For MD17/QM9 dataset, we set an upper bound (6000/1000) for the number of epoches and use an early stop strategy which finishes training if the validation score does not increase after 500/50 epoches. We utilize Adam optimizer and ReduceLROnPlateau learning rate scheduler to optimize models.

**Model hyperparameter tuning.** Hyperparameters were selected to minimize l1 loss on the validation sets. For MD17/QM9, we fix the initial lr to $1e - 3/3e - 4$, batch size to $16/64$, hidden dimension to $256$. The cutoff radius is selected from $[4, 12]$. The number of message passing layer is selected from $[4, 8]$. The dimension of rbf is selected from $[32, 96]$. Please refer to our code for the detailed settings.

**Dataset split.** We randomly split the molecule set into train/validation/test sets. For MD17, the size of the train and validation set are $950, 50$ respectively. All remaining data is used for test. For QM9: The sizes of randomly splited train/validation/test sets are $110000, 10000, 10831$ respectively.

## E  Scalability

To assess the scalability of our model, we show the inference time of random MD17 batches of 32 molecules on an NVIDIA RTX 3090 GPU. The results are shown in Table 4. Note that GemNet consumes too much memory, and only batches of 16 molecules can fit in the GPU. Our model only takes $30\%$ time and $12\%$ space compared with the fastest baseline. Moreover, NoShare use $260\%$ more space and $67\%$ more computation time than GNN-LF with filter sharing technique.

We also compare computational overhead on tasks other than PES in the QM9 dataset (see Table 5). As we use the same model for different tasks in the QM9 dataset, models are only compared on $U_0$ task. GNN-LF achieves the lowest GPU memory consumption, competitive inference speed, and model size.

## F  Existing methods using frame

Though some previous works [24–26, 33, 36] also use the term "frame" or designs similar to "frame", they are very different methods from ours.

The primary motivation of our work is to get rid of equivariant representation for higher and provable expressivity, simpler architecture, and better scalability. We only use equivariant representations in the frame generation and projection process. After projection, all the remaining parts of our model only operates on invariant representations. In contrast, existing works [24, 26, 33, 36] still use both equivariant and invariant representations, resulting in extra complexity even after using frame. For example, functions for equivariant and invariant representations still need to be defined separately, and complex operation is needed to mix the information contained in the two kinds of representations. In addition, our model can beat the representative methods of this kind in both accuracy and scalability on potential energy surface prediction tasks.

Other than the different primary motivation, our model has an entirely different architecture from existing ones.

1. Generating frame: ClofNet [26] and LoCS [25] produces a frame for each pair of nodes and use some process not learnable to produce the frame. Both EGNN [33] and GMN [36] use coordinate embeddings which are initialized with coordinates. Luo et al. [24] initializes the frame with zero. Then these models use some schemes to update the frame. Our model uses a novel message passing scheme to produce frames and will not update it, leading to simpler architecture and low computation overhead.

2. Projection: Existing models [24, 26, 33, 36] only project equivariant features onto the frame, while we also use frame-frame projection, which is verified to be critical both experimentally and theoretically.

3. Message passing layer: Existing models [24–26, 33, 36] all use both invariant representation and equivariant features and pass both invariant and equivariant messages, which needs to mix invariant representations and equivariant representations, update invariant representations, and update equivariant representations, while our model only uses invariant representations, resulting in an entirely different and much simpler design with significantly higher scalability.

4. Our designing tricks, including: message passing scheme to produce frame, filter decomposition, and filter sharing, are not used in [24, 26, 33, 36]. Our experiments and ablation study verified their effectiveness.

Furthermore, existing models use different groups to describe symmetry. Luo et al. [24], Kofinas et al. [25], Du et al. [26] design SO(3)-equivariant model, while our model is O(3)-equivariant. We emphasize that this is not a constraint of our model but a requirement of the task. As most target properties of molecules are O(3)-equivariant (including energy and force we aim to predict), our model can fully describe the symmetry.

Our theoretical analysis is also novel. Luo et al. [24], Satorras et al. [33], Huang et al. [36] have no theoretical analysis of expressivity. Du et al. [26]'s analysis is primarily based on the framework of Dym and Maron [37], which is further based on the polynomial approximation and the group representation theory. The conclusion is that a model needs many message passing layers to approximate high-order polynomials and achieve universality. Our theoretical analysis gets rid of polynomial and group representation and provides a much simpler analysis. We also prove that one message passing layer proposed in our paper are enough to be universal.

In summary, although also using "frame", our work is significantly different to any existing work in either method, theory, or task.

**Gauge-equivariant CNNs** Gauge-equivariance methods [38–40] have never been used in the potential energy surface task. The methods seem similar to ours as they also project equivariant representations onto some selected orientations. However, the differences are apparent.

1. Some of these methods are not strictly O(3)-equivariant. For example, the model of de Haan et al. [39] is not strictly equivariant for angle $\neq 2\pi/N$, while our model (and all existing models for potential enerby surface) is strictly O(3)-equivariant. Loss of O(3)-equivariance leads to high sample complexity.

2. Building grid is infeasible for potential energy surface tasks as atoms can move in the whole space. Moreover, the energy prediction must be a smooth function of the coordinates of atoms. Therefore, the space should not be discretized. The model of Suk et al. [40] works on some discrete grid and cannot be used for the molecular force field.

3. Even though Suk et al. [40] seem to achieve strict O(3)-equivariance with high complexity, it only uses the tangent plane's angle and loses some information. Only **one** angle relative to a reference neighbor is used. Such an angle is expressive enough in a **2D** tangent space because the coordinate can be represented as $(r\cos\theta, r\sin\theta)$. However, for molecule in **3D** space, such an angle is not enough(the coordinates can be represented as $(r\cos\theta\sin\phi, r\sin\theta\sin\phi, r\cos\phi)$. The angles in tangent space only provide $\theta$). In contrast, we use the projection on three frame directions, so our model can fully capture the coordinates.

4. Gauge-equivariance methods all use some constained kernels, which needs careful designation. Our model needs **no specially designed kernel** and can directly use the ordinary message passing scheme. Such simple design leads to provable expressivity, simpler architecture, and low time complexity. Our time complexity is $O(Nn)$, while the that of Suk et al. [40] is $O(Nn^2)$, where $N$ is the number of atoms, $n$ is the maximum node degree).

## G   Expressivity with symmetric input

We use the symbol in Equation 5 SchNet's message can be formalized as follows.

$$\chi(\{(s_j, \vec{r}_{ij}, \vec{E}_j)|r_{ij} < r_c\}) = \rho(\sum_{r_{ij} < r_c} \psi(\text{Concatenate}(s_j, r_{ij})). \tag{62}$$

In implementation, GNN-LF has the following form.

$$\chi(\{(s_j, \vec{r}_{ij}, \vec{E}_j)|r_{ij} < r_c\}) = \rho(\sum_{r_{ij} < r_c} \psi(\text{Concatenate}(P_{\vec{E}_i}(\vec{r}_{ij}), P_{\vec{E}_i}(\vec{E}_j), s_j, r_{ij})). \tag{63}$$

Therefore, for all input molecules, GNN-LF is at least as expressive as SchNet.

**Theorem G.1.** $\forall L \in \mathbb{N}^+$, *for all L layer SchNet, there exists a L layer GNN-LF produce the same output for all input molecule.*

*Proof.* Let the following equation denote the $l^{\text{th}}$ layer SchNet.

$$\chi'_l(\{(s^{(l)}_j, \vec{r}_{ij}, \vec{E}_j)|r_{ij} < r_c\}) = \rho'_l(\sum_{r_{ij}<r_c} \psi'_l(\text{Concatenate}(s^{(l-1)}_j, r_{ij})). \tag{64}$$

Let $\rho_l = \rho'_l$, $\psi_l(\text{Concatenate}(P_{\vec{E}_i}(\vec{r}_{ij}), P_{\vec{E}_i}(\vec{E}_j), s_j, r_{ij})) = \psi'_l(\text{Concatenate}(s_j, r_{ij}))$, which neglects the projection input.

Let the $l^{\text{th}}$ layer of GNN-LF have the following form.

$$\chi_l(\{(s^{(l)}_j, \vec{r}_{ij}, \vec{E}_j)|r_{ij} < r_c\}) = \rho_l(\sum_{r_{ij}<r_c} \psi_l(\text{Concatenate}(s^{(l-1)}_j, r_{ij})). \tag{65}$$

This GNN-LF produces the same output as the SchNet.

$\square$

# H How to overcome the frame degeneration problem.

As shown in Theorem 5.1, if the frame is $O(3)$-equivariant, no matter what scheme is used, the frame will degenerate when the input molecule is symmetric. In other words, the projection degeneration problem roots in the symmetry of molecule. Therefore, we try to break the symmetry by assigning node identity features $s'$ to atoms. The $i^{\text{th}}$ row of $s'$ is $i$. We concatenate $s$ and $s'$ as the new node feature $\tilde{s} \in \mathbb{R}^{N \times (F+1)}$. Let $\eta$ denote a function concatenating node feature $s$ and node identity features $s'$, $\eta(s) = \tilde{s}$. Its inverse function removes the node identity $\eta^{-1}(\tilde{s}) = s$.

In this section, we assume that the cutoff radius is large enough so that local environments cover the whole molecule. Let $[n]$ denote the sequence $1, 2, ..., n$. Let $s \in \mathbb{R}^{N \times F}$ denote the invariant atomic features, $\vec{r} \in \mathbb{R}^{N \times 3}$ denote the 3D coordinates of atoms, and $\vec{r}_i$, the $i^{\text{th}}$ row of $\vec{r}$, denote the coordinates of atom $i$. Let $\vec{r} - \vec{r}_i$ denote an $N \times 3$ matrix whose $j^{\text{th}}$ row is $\vec{r}_j - \vec{r}_i$. We assume that $N > 1$ throughout this section.

Now each atom in the molecule has a different feature. The frame generation is much simpler now.

**Proposition H.1.** *With node identity features, there exists an $O(3)$-equivariant function mapping the local environment $LE_i = \{(\tilde{s}_i, \vec{r}_{ij})|j \in [N]\}$ to a frame $\vec{E}_i \in \mathbb{R}^{3 \times 3}$, and the first $\text{rank}(\vec{E}_i)$ rows of $\vec{E}_i$ form an orthonormal basis of $\text{span}(\{\vec{r}_{ij}|j \in [N]\})$ while other rows are zero.*

*Proof.* For node $i$, we can use the following procedure to produce a frame.

Initialize $\vec{E}_i$ as an empty matrix. For j in $[1, 2, 3, ..., n]$, if $\vec{r}_{ij}$ is linearly independent to row vectors in $\vec{E}_i$, add $\vec{r}_{ij}$ as a row vector of $\vec{E}_i$.

Therefore, when the procedure finishes, the row vectors of $\vec{E}$ form a maximal linearly independent system of $\{\vec{r}_{ij}|j \in [N]\}$.

Then, we use the Gram-Schmidt process to orthonormalize the non-empty row vectors in $\vec{E}$, and then use zero to fill the empty rows in $\vec{E}$ to form a $3 \times 3$ matrix. Therefore, the first $\text{rank}(\vec{E}_i)$ rows of $\vec{E}_i$ are orthonormal, and can linearly express all vector in $\{\vec{r}_{ij}|j \in [N]\}$. In other words, the first $\text{rank}(\vec{E}_i)$ rows of $\vec{E}_i$ form an orthonormal basis of $\text{span}(\{\vec{r}_{ij}|j \in [N]\})$.

Note that $\vec{r}_{ij}$, $\vec{0}$ are $O(3)$-equivariant vectors. Therefore, the frame produced with this scheme is $O(3)$-equivariant. $\square$

With the frame, GNN-LF has the universal approximation property.

**Proposition H.2.** *Assuming that the node identity features are used, and the frame is produced by the method in Proposition H.1. For all $O(3)$-invariant and translation-invariant functions $f(s, \vec{r})$, $f$ can be written as a function of the embeddings of node 1 produced by one message passing layer proposed in Theorem 4.1.*

*Proof.* Let $e_r \in \mathbb{R}^{3 \times 3}$ denote a diagonal matrix whose first $r$ diagonal elements are 1 and others are 0.

With node identity features and method in Proposition H.1, the first $\text{rank}(\vec{E}_i)$ rows of $\vec{E}_i$ form an orthonormal basis of $\text{span}(\{\vec{r}_{ij}|j \in [N]\})$ while other rows are zero. Especial, all vectors in $\{\vec{r}_{1j}|j \in [N]\}$ can be written as linear combination of rows in $\vec{E}_1$, $\vec{r}_{1j} = w_{1j}\vec{E}_j$. Therefore, the projection operation $P_{\vec{E}_1} : \{\vec{r}_{1j}|j \in [N]\} \to \{(\vec{r}_{1j})\vec{E}_1^T|j \in [N]\}$ is injective, as

$$P_{\vec{E}_1}(\vec{r}_{1j})\vec{E}_1 = w_{1j}\vec{E}_1\vec{E}_1^T\vec{E}_1 = w_{1j}e_{\text{rank}(\vec{E}_1)}\vec{E}_1 = w_{1j}\vec{E}_1 = \vec{r}_{1j}. \tag{66}$$

According to the proof of Theorem 4.1, there exists injective function $\chi$ so that node embeddings $z_1 = \chi(\{\tilde{s}_j, P_{\vec{E}_1}(\vec{r}_{1j})|j \in [N]\})$. Note that both $\vec{E}_1$ and $z_1$ are functions of $LE_1$.

Let $\tau$ denote a function $(z_1, \vec{E}_1) = \tau(\{(\tilde{s}_j, \vec{r}_{1j})|j \in [N]\})$, so that

$$\forall o \in O(3), (z_1, \vec{E}_1 o^T) = \tau(\{(\tilde{s}_j, \vec{r}_{1j}o^T)|j \in [N]\}). \tag{67}$$

Moreover, $\tau$ is also an invertible function because

$$\{(\tilde{s}_j, \vec{r}_{1j})|j \in [N]\} = \{(s, p\vec{E}_1)|(s, p) \in \chi^{-1}(z_1)\}. \tag{68}$$

Because the last column of $\tilde{s}$ is the node identity feature, there exists an bijective function $\varphi$ converting set of features to matrix of features.

$$\varphi(\{(\tilde{s}_j, \vec{r}_{1j})|j \in [N]\}) = (s, -(\vec{r} - \vec{r}_1)). \tag{69}$$

Intuitively, it puts the features of atom with node identity $i$ to the $i^{\text{th}}$ row of feature matrix. Similarly,

$$\varphi'(\{(\tilde{s}_j, P_{\vec{E}_1}(\vec{r}_{1j}))|j \in [N]\}) = (s, -(\vec{r} - \vec{r}_1)\vec{E}_1^T). \tag{70}$$

As $f$ is a translation- and O(3)-invariant function,

$$f(s, \vec{r}) = f(s, \vec{r} - \vec{r}_1) = f(s, -(\vec{r} - \vec{r}_1)) = f(\varphi(\{(\tilde{s}_j, \vec{r}_{1j})|j \in [N]\})). \tag{71}$$

Let $g = f \circ \varphi \circ \tau^{-1}$. So

$$g(\vec{E}_1, z_1) = f(s, \vec{r}), \tag{72}$$

$$\forall o \in O(3), g(\vec{E}_1 o^T, z_1) = f(s, \vec{r}o^T) = f(s, \vec{r}). \tag{73}$$

Let $\text{extend}(E_i) \in O(3)$ denote any matrix whose first $\text{rank}(E_i)$ rows equals to $E_i$'s first rows. Therefore,

$$f(s, \vec{r}) = f(s, \vec{r}\text{extend}(\vec{E}_1)^T) = g(\vec{E}_1\text{extend}(\vec{E}_1)^T, z_1) = g(e_{\text{rank}(\vec{E}_1)}, z_1) = g'(\text{rank}(\vec{E}_1), z_1). \tag{74}$$

Note that $\text{rank}(\vec{E}_1) = \text{rank}(\vec{r} - \vec{r}_1) = \text{rank}(P_{\vec{E}_1}(\vec{r} - \vec{r}_1)) = \text{rank}(\iota \circ \varphi' \circ \chi^{-1}(z_i))$, where $\iota$ is a selection function: $\iota(z, -(\vec{r} - \vec{r}_0)\vec{E}_1^T) = -(\vec{r} - \vec{r}_0)\vec{E}_1^T$. Therefore, $f(s, \vec{r}) = g'(\text{rank}(\iota \circ \chi^{-1}(z_1)), z_1) = g''(z_1)$. □

For simplicity, let function $\psi$ denote GNN-LF with node identity features (including adding node identity feature, generating frame, and a message passing layer proposed in Theorem 4.1), $\psi(z, \vec{r})$ is the embeddings of node 1.

Node identity features help avoiding expressivity loss caused by frame degeneration. However, GNN-LF's output is no longer permutation invariant. Therefore, we use the relational pooling method [28], which introduces extra computation overhead and keeps the permutation invariance.

To illustrate this method, we first define some concepts. Function $\pi : [n] \to [n]$ is a permutation iff it is bijective. All permutation on $[n]$ forms the permutation group $S_n$. We compute the output of all possible atom permutations and average them, in order to keep permutation invariance. We define the permutation of matrix here: for all matrix of shape $N \times m$, $\forall \pi \in S_N$, the $i^{\text{th}}$ row of $\pi(M)$ equals to the $(\pi^{-1}(i))^{\text{th}}$ row of $M$.

**Proposition H.3.** *For all $O(3)$-invariant, permutation-invariant and translation invariant function $f(s, \vec{r})$, there exists GNN-LF $\psi$ and some function $g$, with which $\frac{1}{N!} \sum_{\pi \in S_N} g(\psi(\pi(s), \pi(\vec{r})))$ is permutation invariant and equals to $f(s, \vec{r})$.*

*Proof.* Define a "frame" (defined in Definition 1 in [28]) $F : V \to 2^{S_n}$, where $V$ is the embedding space. $\forall v \in V, F(v) = S_n$. So the relational pooling of GNN-LF with node identity features $\langle g \circ \psi \rangle_F(s, \vec{r}) = \frac{1}{N!} \sum_{\pi \in S_N} g(\psi(\pi(s)), \pi(\vec{r}))$. Note that the permutation operation $\pi$ and O(3) operation $o$ commute: $\pi(\vec{r}o^T) = \pi(\vec{r})o^T$. According to Theorem 2 in [28], $\langle g \circ \psi \rangle_F$ is permutation invariant.

According to Theorem 4 in [28], if there exist function $g'$ and $\psi'$ so that $g' \circ \psi' = f$ (the existence is shown in Proposition H.2), there will also exist GNN-LF $\psi$ and function $g$, so that $\langle g \circ \psi \rangle_F(s, \vec{r}) = f(s, \vec{r})$. $\qquad \square$

Therefore, we can completely solve the frame degeneration problem with the relational pooling trick and node identity features. However, the time complexity is up to $O(N!N^2)$, so we only analyze this method theoretically.

## I  Why is global frame more likely to degenerate than local frame?

Let $[N]$ denote the sequence $1, 2, ..., N$. $N$ is the number of atoms in the molecule.

We first consider when local frame degenerates. As shown in Theorem 5.1, the degeneration happens if and only if the local environment is symmetric under some orthogonal transformations.

$$\text{rank}(\vec{E}_i) < 3 \Leftrightarrow \exists o \in O(3), o \neq I, \{(s_i, \vec{r}_{ij}o^T) | r_{ij} < r_c\} = \{(s_i, \vec{r}_{ij}) | r_{ij} < r_c\}. \quad (75)$$

The global frame has the following form,

$$\vec{E} = \sum_{i=1}^{N} \vec{E}_i. \quad (76)$$

We first prove some properties of $\vec{E}$ function.

**Proposition I.1.** *$\vec{E}$ is an $O(3)$-equivariant, translation-invariant, and permutation-invariant function.*

*Proof.* O(3)-equivariance: $\forall o \in O(3), \vec{E}_i(s, \vec{r}o^T) = \vec{E}_i(s, \vec{r})o^T$. Therefore,

$$\vec{E}(s, \vec{r}o^T) = \sum_{i=1}^{N} \vec{E}_i(s, \vec{r}o^T) = (\sum_{i=1}^{N} \vec{E}_i(s, \vec{r}))o^T = \vec{E}(s, \vec{r})o^T. \quad (77)$$

Translation-invariance: For all translation $\vec{t} \in \mathbb{R}^3$, let $\vec{r} + \vec{t}$ denote a matrix of shape $N \times 3$ whose $i^{\text{th}}$ row is $\vec{r}_i + \vec{t}$. As $\vec{E}_i$ is a function of $\vec{r}_i - \vec{r}_j = \vec{r}_i + \vec{t} - (\vec{r}_j + \vec{t})$, $\vec{E}_i(z, \vec{r} + t) = \vec{E}_i(z, \vec{r})$. Therefore,

$$\vec{E}(s, \vec{r} + t) = \sum_{i=1}^{N} \vec{E}_i(s, \vec{r} + t) = \sum_{i=1}^{N} \vec{E}_i(s, \vec{r}) = \vec{E}(s, \vec{r}). \quad (78)$$

Permutation-invariance: for all permutation $\pi \in S_n$, $\pi(\vec{r})_i = \pi(\vec{r})_{\pi^{-1}(i)}$.

$$\vec{E}(\pi(s), \pi(\vec{r})) = \sum_{i=1}^{N} \vec{E}_i(\pi(s), \pi(\vec{r})) \tag{79}$$

$$= \sum_{i=1}^{N} \vec{E}_i(\{(\pi(s)_j, \pi(\vec{r})_i - \pi(\vec{r})_j| \, |\pi(\vec{r})_i - \pi(\vec{r})_j| < r_c\}) \tag{80}$$

$$= \sum_{i=1}^{N} \vec{E}_i(\{(s_{\pi^{-1}(j)}, \vec{r}_{\pi^{-1}(i)} - \vec{r}_{\pi^{-1}(j)})|r_{\pi^{-1}(i)\pi^{-1}(j)} < r_c\}) \tag{81}$$

$$= \sum_{i=1}^{N} \vec{E}_i(\{(s_j, \vec{r}_i - \vec{r}_j|r_{ij} < r_c\}) \tag{82}$$

$$= \vec{E}(s, \vec{r}). \tag{83}$$

$\square$

Then we prove a sufficient condition for global frame degeneration.

**Proposition I.2.** *$rank(\vec{E} < 3)$ if there exists $\vec{t} \in \mathbb{R}^3$ and $o \in O(3), o \neq I$ such that $\{(s_i, \vec{r}_i - \vec{t})|i \in [N]\} = \{(s_i, (\vec{r}_i - \vec{t})o^T)|i \in [N]\}$.*

*Proof.* As $\vec{E}$ is a permutation invariant function,

$$\vec{E} = \vec{E}(\{(s_i, \vec{r}_i)|i \in [N]\}). \tag{84}$$

As $\vec{E}$ is a translation-invariant and $O(3)$-equivariant function,

$$\vec{E}(\{(s_i, (\vec{r}_i - \vec{t})o^T)|i \in [N]\}) = \vec{E}(\{(s_i, (\vec{r}_i - \vec{t}))|i \in [N]\})o^T = \vec{E}(\{(s_i, \vec{r}_i))|i \in [N]\})o^T. \tag{85}$$

Therefore, under the condition $\{(s_i, \vec{r}_i - \vec{t})|i \in [N]\} = \{(s_i, (\vec{r}_i - \vec{t})o^T)|i \in [N]\}$, we have

$$\vec{E}(\{(s_i, \vec{r}_i))|i \in [N]\})o^T = \vec{E}(\{(s_i, \vec{r}_i))|i \in [N]\}), \tag{86}$$

$$\implies \vec{E}(\{(s_i, \vec{r}_i))|i \in [N]\})(I - o^T) = 0. \tag{87}$$

Therefore, $rank(\vec{E}) + rank(I - o^T) - 3 \leq 0$. As $I \neq o^T$, $rank(I - o^T) > 0$, $rank(\vec{E}) < 3$. $\square$

The main difference between the degeneration conditions is the choice of origin. The local frame of atom $i$ degenerates when the molecule is symmetric with atom $i$ as the origin point, while the global frame degenerates if the molecule is symmetric with **any origin point**. Therefore, the global frame is more likely to degenerate.

**Corollary I.1.** *Assume the cutoff radius is large enough so that local environments contain all atoms. If there exists $i$, $rank(\vec{E}_i) < 3$, then $rank(\vec{E}) < 3$.*

*Proof.* As $rank(\vec{E}_i) < 3$, $\exists o \in O(3), o \neq I, \{(s_j, (\vec{r}_i - \vec{r}_j)o^T)|j \in [N]\} = \{(s_j, (\vec{r}_i - \vec{r}_j))|j \in [N]\}$.

Therefore,

$$\{(s_j, -(\vec{r}_i - \vec{r}_j)o^T)|j \in [N]\} = \{(s_j, -(\vec{r}_i - \vec{r}_j))|j \in [N]\} \tag{88}$$

$$\implies \{(s_j, (\vec{r}_j - \vec{r}_i)o^T)|j \in [N]\} = \{(s_j, \vec{r}_j - \vec{r}_i))|j \in [N]\} \tag{89}$$

Let $\vec{t} = \vec{r}_i$, according to Proposition I.2, $rank(\vec{E}) < 3$. $\square$

Therefore, when the cutoff radius is large enough, the global frame will also degenerate if some local frame degenerates.

## J How does GNN-LF keep O(3)-invariance.

The input of GNN-LF is atomic numbers $z \in \mathbb{Z}^N$ and 3D coordinates $\vec{r} \in \mathbb{R}^{N \times 3}$, where $N$ is the number of atoms in our molecule. The energy prediction produced by GNN-LF should be O(3)-equivariant. To formalize, $\forall o \in O(3)$, GNN-LF$(z, \vec{r}) =$ GNN-LF$(z, \vec{r}o^T)$. For example, when the input molecule rotates, the output of GNN-LF should not change.

We state the Definition 2.2 and Lemma 2.1 here again.

**Definition J.1.** *Representation $s$ is called an **invariant representation** if $s(z, \vec{r}) = s(z, \vec{r}o^T), \forall o \in O(3), z \in \mathbb{Z}^N, \vec{r} \in \mathbb{R}^{N \times 3}$. Representation $\vec{v}$ is called an **equivariant representation** if $\vec{v}(z, \vec{r})o^T = \vec{v}(z, \vec{r}o^T), \forall o \in O(3), z \in \mathbb{Z}^N, \vec{r} \in \mathbb{R}^{N \times 3}$.*

**Lemma J.1.**

1. *Any function of invariant representation $s$ will produce an invariant representation.*

2. *Let $s \in \mathbb{R}^F$ denote an invariant representation, $\vec{v} \in \mathbb{R}^{F \times 3}$ denote an equivariant representation. We define $s \cdot \vec{v} \in \mathbb{R}^{F \times 3}$ as a matrix whose $(i, j)$th element is $s_i \vec{v}_{ij}$. When $\vec{v} \in \mathbb{R}^{1 \times 3}$, we first broadcast it along the first dimension. Then the output is also an equivariant representation.*

3. *Let $\vec{v} \in \mathbb{R}^{F \times 3}$ denote an equivariant representation. $\vec{E} \in \mathbb{R}^{3 \times 3}$ denotes an equivariant frame. The projection of $\vec{v}$ to $\vec{E}$, denoted as $P_{\vec{E}}(\vec{v}) := \vec{v}\vec{E}^T$, is an invariant representation in $\mathbb{R}^{F \times 3}$. For $\vec{v}$, $P_{\vec{E}}$ is a bijective function. Its inverse $P_{\vec{E}}^{-1}$ convert an invariant representation $s \in \mathbb{R}^{F \times 3}$ to an equivariant representation in $\mathbb{R}^{F \times 3}$, $P_{\vec{E}}^{-1}(s) = s\vec{E}$.*

4. *Projection of $\vec{v}$ to a general equivariant representation $\vec{v}' \in \mathbb{R}^{F' \times 3}$ can also be defined. It produces an invariant representation in $\mathbb{R}^{F \times F'}$, $P_{\vec{v}'}(\vec{v}) = \vec{v}\vec{v}'^T$.*

As shown in Figure 1, GNN-LF first generates a frame for each atom and projects equivariant features of neighbor atoms onto the frame. A graph with only invariant features is then produced. An ordinary GNN is then used to process the graph and produce the output. We illustate them step by step.

**Notations.** The initial node feature of node $i$, $z_i \in \mathbb{N}$, is an integer atomic number, which neural network cannot process directly. So we first use an embedding layer to transform $z_i$ to float features $s_i = s(z_i) \in \mathbb{R}^F$, where $F$ is the hidden dimension. According to the first point of Lemma J.1, $s_i$ is an invariant representation.

$\vec{r}_i \in \mathbb{R}^{1 \times 3}$, the 3D coordinates of atom $i$, is an equivariant representation. $\vec{r}_{ij} = \vec{r}_i - \vec{r}_j \in \mathbb{R}^{1 \times 3}$ is the position of atom $i$ relative to atom $j$.

$$\forall o \in O(3), \vec{r}_{ij}(z, \vec{r}o^T) = \vec{r}_i o^T - \vec{r}_j o^T = \vec{r}_{ij}(z, \vec{r})o^T, \tag{90}$$

so $\vec{r}_{ij}$ is an equivariant representation. $r_{ij}$ denotes the distance between atom $i$ and atom $j$. $r_{ij} = \sqrt{\vec{r}_{ij}\vec{r}_{ij}^T} \in \mathbb{R}$. According to the fourth point of Lemma J.1, $\vec{r}_{ij}\vec{r}_{ij}^T$ is an invariant representation. According to the first point of Lemma J.1, $r_{ij} = \sqrt{\vec{r}_{ij}\vec{r}_{ij}^T}$ is thus an invariant representation.

**Frame Generation.** As shown in Equation 8, our frame has the following form.

$$\vec{E}_i = \sum_{j \neq i, r_{ij} < r_c} \frac{w(r_{ij})}{r_{ij}}(f(r_{ij}) \odot s_j) \cdot \vec{r}_{ij}, \tag{91}$$

where $w(r_{ij}) \in \mathbb{R}$ and $f(r_{ij}) \in \mathbb{R}^F$ denotes two function of $r_{ij}$, $\odot$ denotes Hadamard product. $\frac{w(r_{ij})}{r_{ij}}(f(r_{ij}) \odot s_j)$ as a whole is a function of $r_{ij}$ and $s_j$, which are both invariant representations. According to the first point of Lemma J.1, $\frac{w(r_{ij})}{r_{ij}}(f(r_{ij}) \odot s_j)$ is an invariant representation $\in \mathbb{R}^F$. $\cdot$ denotes the scale operation described in the second point of Lemma J.1, so $\frac{w(r_{ij})}{r_{ij}}(f(r_{ij}) \odot s_j) \cdot \vec{r}_{ij}$ is an equivariant representation. The frame of atom $i$, namely $\vec{E}_i \in \mathbb{R}^{F \times 3}$, is an equivariant

**Table 6:** Mean average error on the MD17 dataset. Units: energy ($\mathcal{E}$) (kcal/mol) , forces ($\mathcal{F}$) (kcal/mol/Å).Tuned means GNN-LF with tuned cutoff radius. cf* means GNN-LF with cutoff *Å. Torchmd is the strongest baseline.

|  |  | cf3.5 | cf4.5 | cf5.5 | cf6.5 | cf7.5 | cf8.5 | cf9.5 | Tuned | Torchmd |
|---|---|---|---|---|---|---|---|---|---|---|
| Aspirin | $\mathcal{E}$ | 0.1544 | 0.9091 | 0.1378 | 0.1896 | 0.1322 | 0.1312 | 0.1312 | 0.1342 | 0.1240 |
|  | $\mathcal{F}$ | 0.3092 | 1.6694 | 0.2164 | 0.4170 | 0.1896 | 0.1954 | 0.1954 | 0.2018 | 0.2550 |
| Benzene | $\mathcal{E}$ | 0.0686 | 0.0701 | 0.0696 | 0.0689 | 0.0690 | 0.0694 | 0.0697 | 0.0686 | 0.0560 |
|  | $\mathcal{F}$ | 0.1559 | 0.1490 | 0.1624 | 0.1489 | 0.1496 | 0.1492 | 0.1489 | 0.1506 | 0.2010 |
| Ethanol | $\mathcal{E}$ | 0.0516 | 0.0519 | 0.0523 | 0.0514 | 0.0514 | 0.0514 | 0.0514 | 0.0520 | 0.0540 |
|  | $\mathcal{F}$ | 0.0874 | 0.0885 | 0.0877 | 0.0798 | 0.0798 | 0.0798 | 0.0798 | 0.0814 | 0.1160 |
| Malonaldehyde | $\mathcal{E}$ | 0.0772 | 0.0780 | 0.0784 | 0.0744 | 0.0744 | 0.0747 | 0.0747 | 0.0764 | 0.0790 |
|  | $\mathcal{F}$ | 0.1622 | 0.1623 | 0.1631 | 0.1190 | 0.1128 | 0.1126 | 0.1126 | 0.1259 | 0.1760 |
| Naphthalene | $\mathcal{E}$ | 0.1153 | 0.1153 | 0.1590 | 0.1148 | 0.1124 | 0.1124 | 0.1124 | 0.1136 | 0.0850 |
|  | $\mathcal{F}$ | 0.0538 | 0.0538 | 0.1261 | 0.0506 | 0.0507 | 0.0507 | 0.0507 | 0.0550 | 0.0600 |
| Salicylic | $\mathcal{E}$ | 0.1110 | 0.1238 | 0.1082 | 0.1090 | 0.1077 | 0.1078 | 0.1085 | 0.1081 | 0.0940 |
|  | $\mathcal{F}$ | 0.1335 | 0.1525 | 0.1037 | 0.1019 | 0.1021 | 0.1014 | 0.1013 | 0.1005 | 0.1350 |
| Toluene | $\mathcal{E}$ | 0.0947 | 0.1004 | 0.1601 | 0.0924 | 0.0924 | 0.0924 | 0.0924 | 0.0930 | 0.0740 |
|  | $\mathcal{F}$ | 0.0662 | 0.0664 | 0.2502 | 0.0518 | 0.0518 | 0.0518 | 0.0518 | 0.0543 | 0.0660 |
| Uracil | $\mathcal{E}$ | 58.794 | 0.149 | 0.1037 | 0.2334 | 0.1038 | 0.1039 | 0.1037 | 0.1037 | 0.0960 |
|  | $\mathcal{F}$ | 17.0794 | 0.1106 | 0.077 | 0.4496 | 0.0842 | 0.0771 | 0.077 | 0.0751 | 0.0940 |

representation, because

$$\vec{E}_i(z, \vec{r}o^T) = \sum_{j \neq i, r_{ij} < r_c} (\frac{w(r_{ij})}{r_{ij}} (f(r_{ij}) \odot s_j) \cdot \vec{r}_{ij}o^T), \tag{92}$$

$$= (\sum_{j \neq i, r_{ij} < r_c} \frac{w(r_{ij})}{r_{ij}} (f(r_{ij}) \odot s_j) \cdot \vec{r}_{ij})o^T = \vec{E}_i(z, \vec{r})o^T. \tag{93}$$

**Projection.** Projection is composed of two parts. As shown in Equation 9 and Equation 6.

$$d_{ij}^1 = \frac{1}{r_{ij}} (\vec{r}_{ij} \vec{E}_i^T) d_{ij}^2 = \text{diag}(W_1 \vec{E}_j \vec{E}_i^T W_2^T), \tag{94}$$

where $W_1, W_2 \in \mathbb{R}^{F \times F}$ are two learnable linear layers. According to the fourth point of lemma J.1, $d_{ij}^1 = \vec{r}_{ij} \vec{E}_i^T$ are invariant representations. According to the fourth point of lemma J.1, $\vec{E}_j \vec{E}_i^T$ are invariant representations. $d_{ij}^2 = \text{diag}(W_1 \vec{E}_j \vec{E}_i^T W_2^T)$ is a function of $\vec{E}_j \vec{E}_i^T$, so $d_{ij}^2$ are invariant representations.

**Graph Neural Network.** We use an ordinary GNN to produce the energy prediction. The GNN takes $s_i$ as the input node features and $(r_{ij}, d_{ij}^1, d_{ij}^2)$ as the input edge features.

$$\text{GNN-LF}(z, \vec{r}) = \text{GNN}(\{s_i | i = 1, 2, .., N\}, \{(r_{ij}, d_{ij}^1, d_{ij}^2) | i = 1, 2, .., N, j = 1, 2, .., N\}). \tag{95}$$

As all inputs of GNN is invariant to O(3) operation, the energy prediction will also be O(3)-invariant.

Our GNN has an ordinary message passing scheme. The message from atom $j$ to atom $r$ is

$$m_{ij} = f_2(r_{ij}, d_{ij}^1, d_{ij}^2) \odot s_j, \tag{96}$$

where $f'$ is a neural network, whose output $\in \mathbb{R}^F$. The message combines the features of edge $i, j$ and node $j$. Each message passing layer will update the node feature $s_i$.

$$s_i \leftarrow s_i + g(\sum_{j \in N(i)} m_{ij}), \tag{97}$$

where $g$ is a multi-layer perceptron, $N(i)$ is the set of neighbor nodes of node $i$.

After some message passing processes, $s_i$ contains rich graph information. The energy prediction is

$$\hat{E} = h(\sum_{i=1}^N s_i), \tag{98}$$

where $h$ is a multi-layer perceptron.

**Table 7:** Results on MD17 with different splits. Units: energy ($\mathcal{E}$) (kcal/mol) , forces ($\mathcal{F}$) (kcal/mol/Å).

| Molecule | Target | Our split | DimeNet split |
|---|---|---|---|
| Aspirin | $\mathcal{E}$ | 0.1342 | 0.1294 |
| | $\mathcal{F}$ | 0.2018 | 0.1902 |
| Benzene | $\mathcal{E}$ | 0.0686 | 0.0695 |
| | $\mathcal{F}$ | 0.1506 | 0.1477 |
| Ethanol | $\mathcal{E}$ | 0.052 | 0.051 |
| | $\mathcal{F}$ | 0.0814 | 0.078 |
| Malonaldehyde | $\mathcal{E}$ | 0.0764 | 0.074 |
| | $\mathcal{F}$ | 0.1259 | 0.1147 |
| Naphthalene | $\mathcal{E}$ | 0.1136 | 0.1138 |
| | $\mathcal{F}$ | 0.055 | 0.0493 |
| Salicylic acid | $\mathcal{E}$ | 0.1081 | 0.1072 |
| | $\mathcal{F}$ | 0.1005 | 0.097 |
| Toluene | $\mathcal{E}$ | 0.093 | 0.0914 |
| | $\mathcal{F}$ | 0.0543 | 0.0499 |
| Uracil | $\mathcal{E}$ | 0.1037 | 0.1033 |
| | $\mathcal{F}$ | 0.0751 | 0.0763 |

## K  How cutoff radius affects performance

Instead of taking a physically motivated cutoff radius, we set it to be a hyperparameter and tune it. Intensive hyperparameter tuning may prohibit GNN-LF from real-world applications. However, we find that GNN-LF is robust to the cutoff radius and does not need a lot of tuning.

As shown in Table 6, when the cutoff radius is low, the accuracy is low and unstable. However, when the cutoff radius is large enough, GNN-LF outperforms the strongest baseline torchmd and achieves the performance of GNN-LF with a tuned cutoff radius.

## L  Results with DimeNet split

Our baselines take slightly different dataset splits. For comparison, we use the same split as our strongest baseline [10]. It is also the split with the fewest training and validation samples and, thus, the most challenging setting. Other baselines may use slightly larger training and validation datasets. For example, in the MD17 dataset, our split uses 950 training samples, while DimeNet uses 1000 training samples. With the split of DimeNet, the performance of GNN-LF increases by 0.5% on average (see Table 7). So the differences in dataset split will not hamper our conclusion: GNN-LF achieves state-of-the-art performance in PES tasks.

## M  Gap between GNN-LF's implementation and expressivity analysis

There are three gaps between implementation and expressivity analysis. We clarify them as follows.

### M.1  Frame ensemble

The theoretical expressivity can be ensured if the frame contains at least 3 vectors. So the choice of $F$, the number of vectors in the frame, is mainly for implementation convenience. We set $F$=hidden dimension, so GNN-LF does not need an extra linear layer to change representations' dimension when producing frames.

We also do ablation study. Let 1-frame denote GNN-LF with a single frame. Though frame ensemble contributes significant performance gain, GNN-LF with only one frame still achieves the second-best performance among existing models.

The experimental results in MD17 dataset are shown in Table 8. Ablation of frame ensemble leads to 10% test loss increase. However, the performance of 1-frame is still competitive, as 1-frame

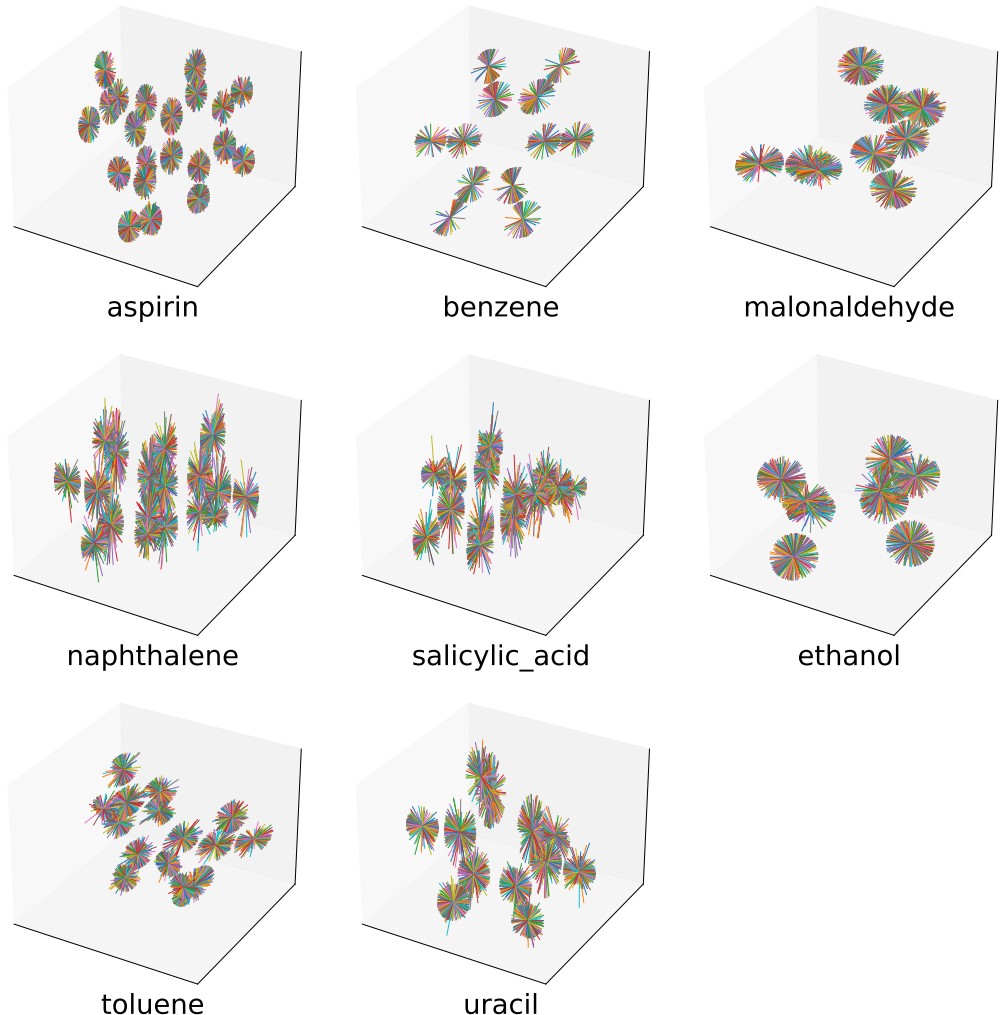

**Figure 5:** Visualization of frames in randomly selected molecules in MD17 dataset. Here frame vectors are represented as lines rooted in atoms.

outperforms all baselines on $3/16$ targets and achieves the second-best performance on $11/16$ targets. The outstanding performance of GNN-LF validates our expressivity analysis.

Though frame ensemble is not vital for performance, we always use it in GNN-LF. As GNN-LF generates frames and projections only once, using frame ensembles will not lead to significant computation overhead. In the setting of Table 4, both 1-frame and GNN-LF with frame ensemble take 9 ms per inference iteration.

## M.2 Orthogonality Constraint

Let $f$ denote the target function. In expressivity analysis, we have shown that there exists parameter $w$, GNN-LF with $w$ can approximate $f$, and the frames produced by GNN-LF are orthogonal. Then this conclusion $\Rightarrow$ there exists parameter $w$, GNN-LF with $w$ can approximate $f$

$\Leftrightarrow$ GNN-LF without the constraint can approximate $f$ Therefore, without the constraint, GNN-LF can still maintain expressivity.

In other words, producing orthogonal frames is one particular solution to our task. By building a specific solution, we prove the existence of the solution. In implementation, optimized GNN-LF

**Table 8:** Results on the MD17 dataset. Units: energy ($\mathcal{E}$) (kcal/mol) and forces ($\mathcal{F}$) (kcal/mol/Å).

| Molecule | Target | 1-frame | DimeNet | GemNet | PaiNN | TorchMD | 1-frame | GNN-LF |
|---|---|---|---|---|---|---|---|---|
| Aspirin | $\mathcal{E}$ | 0.1474 | 0.204 | - | 0.1670 | **0.1240** | 0.1474 | 0.1342 |
| | $\mathcal{F}$ | 0.2784 | 0.499 | **0.2168** | 0.3380 | 0.2550 | 0.2784 | 0.2018 |
| Benzene | $\mathcal{E}$ | 0.0692 | 0.078 | - | - | **0.0560** | 0.0692 | 0.0686 |
| | $\mathcal{F}$ | 0.1532 | 0.187 | **0.1453** | - | 0.2010 | 0.1532 | 0.1506 |
| Ethanol | $\mathcal{E}$ | **0.0525** | 0.064 | - | 0.0640 | 0.0540 | 0.0525 | 0.0520 |
| | $\mathcal{F}$ | 0.0897 | 0.230 | **0.0853** | 0.2240 | 0.1160 | 0.0897 | 0.0814 |
| Malonaldehyde | $\mathcal{E}$ | **0.0789** | 0.104 | - | 0.0910 | 0.0790 | 0.0789 | 0.0764 |
| | $\mathcal{F}$ | 0.1651 | 0.383 | **0.1545** | 0.3190 | 0.1760 | 0.1651 | 0.1259 |
| Naphthalene | $\mathcal{E}$ | 0.1138 | 0.122 | - | 0.1660 | **0.0850** | 0.1138 | 0.1136 |
| | $\mathcal{F}$ | 0.0606 | 0.215 | **0.0553** | 0.0770 | 0.0600 | 0.0606 | 0.0550 |
| Salicylic acid | $\mathcal{E}$ | 0.1088 | 0.134 | - | 0.1660 | 0.0940 | 0.1088 | 0.1081 |
| | $\mathcal{F}$ | 0.1290 | 0.374 | **0.1268** | 0.1950 | 0.1350 | 0.1290 | 0.1005 |
| Toluene | $\mathcal{E}$ | 0.0997 | 0.102 | - | 0.0950 | **0.0740** | 0.0997 | 0.0939 |
| | $\mathcal{F}$ | 0.0682 | 0.216 | **0.0600** | 0.0940 | 0.066 | 0.0682 | 0.0543 |
| Uracil | $\mathcal{E}$ | 0.1048 | 0.115 | - | 0.1060 | **0.096** | 0.1048 | 0.1037 |
| | $\mathcal{F}$ | **0.0944** | 0.301 | 0.0969 | 0.1390 | 0.094 | 0.0944 | 0.0751 |

may produce orthogonal frames, and it can also reach other solutions and thus does not produce orthogonal frames.

As expressivity analysis only considers the existence of a solution and ignores how to reach the solution, it is somehow counter-intuitive. However, the theoretical analysis is still helpful and motivates the designs of GNN-LF.

Moreover, removal of orthogonal constraint does not result in frame degeneration. We visualize local frames of atoms in Figure 5. In these molecules, frame vector directions are diverse. Therefore, frames are not likely to degenerate, and frames in the same ensemble vary greatly rather than collapse into a single frame.

## M.3 Implementation of frame-frame projection

In theory, frame-frame projection is $\vec{E}_i \vec{E}_j^T$. However, in implementation, we use $\mathrm{diag}(W_1 \vec{E}_i \vec{E}_j^T W_2^T)$ (see Equation 10). This section explains the reason for the difference.

Directly using $\vec{E}_i \vec{E}_j^T$ leads to large computation overhead. $\vec{E}_i \vec{E}_j^T$ is a matrix $\in \mathbb{R}^{F \times F}$, where $F$ is the hidden dimension, usually 256. Flattening $\vec{E}_i \vec{E}_j^T$ and transforming it to $F$ dimension needs at least a linear layer with 16M parameters (ten times more than the total number of parameters of GNN-LF in MD17 dataset), which is infeasible. Therefore, sampling elements in $\vec{E}_i \vec{E}_j^T$ is a must.

Moreover, our sampling method will not hamper expressivity. In theory, frames with 3 vectors and $3 \times 3$ frame-frame projections are enough. Therefore, simply selecting a $3 \times 3$ diagonal block in $\vec{E}\vec{E}^T$ can fulfill the theoretical requirements. We use a learnable process to simulate this operation.

Now we explain the sampling method in GNN-LF. Note that we do not directly take the diagonals of $\vec{E}_i \vec{E}_j^T$. Instead, we use $\mathrm{diag}(W_1 \vec{E}_i \vec{E}_j^T W_2^T)$, where $W_1, W_2 \in \mathbb{R}^{F \times F}$ are two learnable matrix used to select elements. Appropriate $W_1, W_2$ can select arbitrary $F$ elements in $\vec{E}_i \vec{E}_j^T$. For example, given

$$W_1 = \sum_{i=1}^{F} 1_{i,a_i}, W_2 = \sum_{i=1}^{F} 1_{i,b_i}, \tag{99}$$

where $1_{i,j}$ denotes the matrix whose $(i,j)$ elements is 1, other elements are 0,

$$\mathrm{diag}(W_1 \vec{E}_i \vec{E}_j^T W_2^T) = [(\vec{E}_i \vec{E}_j^T)_{a_i b_i} | i = 1, 2, .., F]. \tag{100}$$

