# OpenReview forum: "Graph Neural Network with Local Frame for Molecular Potential Energy Surface"
_logconference.io/LOG/2022/Conference — LoG 2022 Poster_

### Official Review · Reviewer_cgRj · 2022-10-20

**Overall Score:** 8
**Confidence:** 4

**Review:**

**Summary**

This work proposes a novel 3D molecular representation learning method in which ordinary GNN with novel local frames (GNN-LF) are used to convert equivariant features to invariant features. The proposed GNN-LF captures the geometric information of molecules and predicts potential energy surface and forces.

**Strong points**

1. Solid theoretical analysis is conducted to show that the proposed coordinate projection and frame-frame projection can help even ordinary GNN to keep expressivity.

2. The experimental results on MD17 and QM9 are very good compared to baseline methods. It strongly supports the high expressivity of the proposed GNN-LF.

3. The reduced inference time and GPU memory consumption are also valuable strengths of this method.

**Weak points**

1. I wonder if the same dataset split is used when comparing with baseline methods. It’s not a fair comparison if dataset splits are not the same. I understand that those baselines may use different splits, but you should follow at least one of them, for example, DimeNet, for a fairer comparison.

2. In addition to existing baselines, could you also compare MAE, number of parameters, inference time and GPU consumption with some recent methods, such as EGNN[1], and ComENet[2].

3. In line 596, it shows that only 11,000 molecules are used in training. I believe it’s just a typo.

This is an interesting paper with solid theoretical analysis and good experimental results. It would be more convincing if my concerns and questions above could be properly addressed.

**Reference**

[1]. Satorras, Vıctor Garcia, Emiel Hoogeboom, and Max Welling. "E (n) equivariant graph neural networks." International conference on machine learning. PMLR, 2021.

[2]. Wang, Limei, et al. "ComENet: Towards Complete and Efficient Message Passing for 3D Molecular Graphs." arXiv preprint arXiv:2206.08515 (2022).

---

### Official Review · Reviewer_Y7yJ · 2022-10-21

**Overall Score:** 6
**Confidence:** 3

**Review:**

Summary:

This paper is motivated to propose a more effective and less complicated graph-based neural network for modeling molecular dynamics. The previous message-passing schemes need special designs to capture geometric information and fulfill symmetry requirements like rotation equivariance. The authors argue that the previous designs are too complicated and not required, so they propose a GNN based on simple local frames, which have simpler architectures than the previous works have. Instead of equivariant features, they propose a local-frame GNN that only uses invariant features. To demonstrate the effectiveness of their model, they theoretically justify the expressivity of their proposed method, and empirically compare their work with classical baselines on multiple benchmark molecular datasets.

Strengths And Weaknesses:

Pros and Cons are discussed in more detail in the following paragraphs.

Strengths:

1. The authors propose a novel local-frame GNN (GNN-LF) for modeling molecular dynamics.

2. The proposed GNN-LF achieves state-of-the-art results on multiple benchmark molecular datasets.

3. The proposed GNN-LF is simpler and sounds more efficient than previous designs but its expressivity is not limited. The expressivity of GNN-LF is theoretically supported.

Weaknesses:

1. The paper is kind of disorganized and hard to follow. The writing and logic can be improved a lot.

2. Some points are not well clarified.

Overall:

I vote for weak acceptance. The authors have a strong motivation to design the novel graph neural network based on local frames (GNN-LF). They state many theories to justify the reason for computing local frames, we see that GNN-LF is a simple design while its expressivity is not limited. Additionally, the authors compare GNN-LF with several classical baselines and SOTA models on multiple benchmark molecular datasets, and the empirical results are solid. The proposed GNN-LF is simpler but stronger in terms of its capability for modeling molecular dynamics. However, the paper is not well organized and not easy to follow, and the quality of the writing could still be improved a lot. Overall, considering their strong results but poor writing, I vote for a weak acceptance.

Major Pros:

1. This is a motivated paper. The authors argue that previous GNNs require special designs to capture geometric information
and fulfill symmetry requirements like rotation equivariance, and such design is too complicated. Thus, they design a simpler model which learns molecular dynamics.

2. GNN-LF is novel. Instead of using equivariant representations, the authors covert equivariant features like 3D coordinates to invariant features. Moreover, by the coordinate projection, ordinary GNNs can encode molecules injectively and reach maximum expressivity to learn molecular dynamics. First, GNN-LF produces a frame equivariant to O(3) transformations.  Then, it projects the coordinate information and neighbor atoms on the frame as the edge features.

3. GNN-LF is well-positioned against existing works. The authors illustrate the difference between GNN-LF and previous models. They also discuss the difference in expressivity and how invariant features can be more useful. Empirically, the authors fairly compare GNN-LF with existing works and show that GNN-LF can achieve state-of-the-art results on multiple molecular datasets.

Major Cons:

1. GNN-LF sounds to be a more efficient model than existing works because the model computes invariant features with an ordinary GNN and does not use GNNs with any special equivariant representation. So I want to see the computation time (including training and inference costs) compared with the baselines in the experimental section.

2. For the frame-frame projection, the authors claim to take the diagonal elements of the projection because $EE^T$ is a large matrix. However, the logic is not clear. I do not understand why $EE^T$ is large so it is better to take the diagonal elements only. In my opinion, it is going to hurt the expressivity to some extent. I hope the authors can justify this point. Moreover, is there any better solution for that?

Limitations:

N/A

---

### Official Review · Reviewer_XK6z · 2022-10-21

**Overall Score:** 8
**Confidence:** 4

**Review:**

### Summary

This paper proposes local frames in geometric graph networks for modelling
molecular potential energy surfaces. Local frames grant equivariance to $\mathrm{O}(3)$ and
decouple the GNN design for the symmetries of the problem, allowing
the use of arbitrary GNN architectures. The authors propose a method to predict local
frames and a GNN architecture  that encodes the local environments in local
frames. Experiments on two well-known datasets, MD17 and QM9, showcase the
effectiveness and efficiency of the proposed method.

### Strengths

The paper is well written and easy to follow; the method sections are well disentangled from one another and allow for a better understanding of individual components.
Similarly, the figures clearly and intuitively illustrate the intended outcomes, namely the illustration of the method,
inadequacies of previous methods, and how symmetries affect the frames.
Experiments on PES and other chemical properties showcase the effectiveness of
the method compared to state-of-the-art methods.
Moreover, meaningful ablation studies successfully probe individual modules of the network and demonstrate the importance of each component.
The paper includes an extensive theoretical analysis of the method and its
expressivity.

### Weaknesses

The mathematical formulations and notation are often not very "clean" and sometimes sloppy.
This makes understanding the derivations hard and quite laborious.
This is especially true in the appendix, where there are often proofs with inline equations.
One example is in lines 471-475.
One suggestion regarding notation is the use of boldface instead of arrow notation, while removing indices when their purpose is clear can enhance clarity.

The comparison to related work on local frame graph networks seems incomplete.
Namely, Luo et al. [1], and Kofinas et al. [2] propose local coordinate frame
graph networks, with almost the same mathematical definition of local frames as this method (with the exception of the reflection symmetry).
Notably, the work of Kofinas et al. [2] is not mentioned at all in related work, even though it shares a lot of similarities with this work.
For example, they use local frames to induce invariance in geometric graphs, and
perform inverse transformations at the output of the network to obtain equivariance.
Thus, they can also use arbitrary GNNs whose design is not bound to the symmetries of the problem.
Further, and uniquely from other works, they also incorporate orientations of neighbouring nodes in their local frames and transform them using rotation matrix representations,
which is very similar to the frame-frame projection of this paper.
In the same spirit, Luo et al. [1] also use local frames in a similar fashion to induce invariance/equivariance, and can also use arbitrary GNNs.

While the missing references are quite important and the authors should consider addressing them in the camera-ready version of the paper,
these two works do indeed differ from the proposed method in a number of ways.
First, the two aforementioned works focus on different tasks (Luo et
al. [1] work with point clouds, and Kofinas et al. [2] work with interacting systems),
and the symmetry groups are different, (SO(3) vs O(3), or SE(3) vs E(3)).
Finally, this work has its own technical contributions (including the estimation
of the local frame), as well as extensive theoretical analysis on the
expressivity of local frame graph networks.

### Suggested Experiments

The estimation of local frames relies on the choice of a cutoff radius. It would
be interesting to see the importance of this variable in the estimation of
local frames. Is the estimation robust to the choice of the
cutoff radius? Qualitatively, would you expect the method to require a lot of hyperparameter tuning in
settings with a heuristic cutoff radius (instead of a physically motivated one)?
On a similar fashion, is the frame estimation and the method overall robust to noisy node positions?

In Luo et al. [1], the frame estimation poses an non-negligible overhead in the
method. Is this also the case for your method?
What is the cost of computing the frame in eq. 6?

Visualizations of the learned frames can provide valuable insights into the
method.
I suggest adding some qualitative examples of them for the
camera-ready version of the paper. Do they correspond to physically meaningful frames?

Your method exhibits excellent performance in molecular datasets with E(3) symmetry.
Is there something that makes it particularly suitable for this setting?
Or do you expect it to perform well on different geometric graph settings, e.g. n-body-problems or point clouds,
and settings with different symmetry groups, e.g. SE(3)?

Does the performance of your method increase even more when you increase the number of parameters, e.g. setting them equal to the bigger models?
Or do you expect it to plateau and have diminishing returns?

### Clarity

I find the description of producing frames in lines 268-271 quite unclear. What is the meaning of $E_i \in \mathbb{R}^{F \times 3}$?
Can it be interpreted as multiple sets of bases for each node, similar to Luo et al. [1] predicting multiple orientations?
Similarly, I do not understand the lemma in lines 90-91, is it used in frame-frame conversion in lines 274-276?

Looking at eq. 6, it is not exactly clear to me how the estimated frames are orthogonal matrices.
Can you elaborate on this?
Furthermore, what is the geometric interpretation of summing the local frames to compute the global frame in eq. 7?
Assuming you perform a standard summation, the global frame matrix is no longer orthogonal and thus, no longer a frame.

In most theorems and propositions, you do not explicitly mention the appendix sections for their proofs.
This makes it hard for the reader to find them and disrupts the flow of the paper.

In definition 2.1, since $h: \mathbb{X} \to \mathbb{Y}$, it would be better to use the representations $\rho^\mathbb{X}[g]$ and $\rho^\mathbb{Y}[g]$ instead of $g$
to avoid confusion.

In multiple occasions, you use $\circ$ (eq. 6, eq. 8, line 84) without explicitly defining it.
It would be better to explicitly define it, remove it, or use the more common notation $\cdot$.

In eq. 4 you use $\varphi$ and $\phi$, while they are practically the same letter, which can confuse the reader.
The same is true for energy $\hat{E}$ and frame $\vec{E}$, both notated with the same letter, albeit in different forms.
I would suggest using a different notation for one of the symbols in each case.

### Additional Notes

Some figures and tables, namely figure 1, table 1 and table 2
have been pushed outside the normal document margins, too close to the header.
Please respect the document margins.

Lines 42 & 209 are too close to figure. Please keep a normal vertical space between text and figures.

Following the conference guidelines, I suggest the use of booktabs for tables and avoiding vertical rules in tables.

### Recommendation

I recommend that this paper is accepted.
It is a well-written paper with important technical contributions and a strong
theoretical analysis on local frame graph networks.
Its clarity could increase in certain places, and the related work could be more thorough.
Additional experiments and ablations could further solidify its novelties.

### References

[1] Shitong Luo, Jiahan Li, Jiaqi Guan, Yufeng Su, Chaoran Cheng, Jian Peng, and Jianzhu Ma. "Equivariant Point Cloud Analysis via Learning Orientations for Message Passing". CVPR 2022.

[2] Miltiadis Kofinas, Naveen Shankar Nagaraja, and Efstratios Gavves. "Roto-translated Local Coordinate Frames for Interacting Dynamical Systems". NeurIPS 2021.

---

### Official Review · Reviewer_KxiD · 2022-10-23

**Overall Score:** 6
**Confidence:** 3

**Review:**

Targeting at less overhead in designing complicated architectures for geometric deep learning and training, the paper proposes a simple yet theoretical-grounded representation of equivariant features, named "frame" -- rather than enforcing equivariance properties in architectures. With frame features, original GNN models can be directly applied with the guaranteed expressiveness as equivariant neural networks.

Advantages:

- Solid theory to support the proposed featurization
- The simplicity nature and effectiveness of frame is appreciated

Disadvantages:

- The presentation can be more structured. I read back and forth for 5+ times before I can get the real idea of frame
- Some incorrect claims. It is said in line 132 "Our model can fully describe symmetry, while existing models cannot". Per my knowledge, equivariant neural networks are capable to capture symmetry (i.e. equivariant), since symmetry would not affect the geometric message passing. i.e. SE(3)-transformer. To enforce mirroring leading to the same energy (as stated in line 133), it is trivial via, cascading with an invariant head. This equivariance should not be the main advantage of the proposed featurization, but efficiency or model parameter numbers (since you don't need complicated models)
- Per my last point, I am confused about the experiment results: why it can achieve the best. I think simplicity is the greatest merit of frame rather than performance, thus it does not make sense to me that it outperforms. Moreover, it would benefit the paper if authors show the time/architecture complexity comparison.
- Missing reference. A lot of equivariant neural network works should be included and compared. E.g. SE(3)-transformer.

---

### Meta-Review · Area_Chair_ANrv · 2022-11-13

**Confidence:** 4
**Recommendation:** Accept

**Meta Review:**

This work studies graph neural networks for molecular dynamics and force prediction. All reviewers are marginally positive about this work after rebuttals and most of review concerns have been addressed during rebuttals. I wish to recommend a conditional accept given the authors are willing to address my below concerns.

1. GNNs for molecular simulation and prediction is not a very new area and there has been quite a bit prior work on 3D GNNs etc. All these are closely related to this work and thus the authors should put this work into proper contexts in terms of discussions and experimental comparisons. Simply stating that this work focuses on potential energy surface as a ground for ignoring related work is not a valid excuse.

2. This work only used two small and saturated datasets for experiments. In the literature, there are other larger and newer datasets that should be used, notably the OCP datasets.

---

### Decision · Program_Chairs · 2022-11-23

**Decision:**

Accept (Poster)

**Comment:**

PCs agree with the AC assessment. We strongly encourage authors to incorporate these suggestions in order to improve clarity, accessibility, and the relation to existing work.